# Phage-assisted evolution of allosteric protein switches

Nicholas T. Southern [1], Anna von Bachmann [1,2], Alisa Hovsepyan[1], Marielouise Griebl [1], Benedict Wolf [1], Nina Lemmen [1], Ann-Sophie Kroell [1], Simon Westermann[1], Jan Mathony [1] ✉ & Dominik Niopek [1] ✉

Allostery, the transmission of locally induced conformational changes to distant functional sites, is a key mechanism for protein regulation. Artificial allosteric effectors enable remote manipulation of cell function; their engineering, however, is hampered by our limited understanding of allosteric residue networks. Here, we introduce a phage-assisted evolution platform for in vivo optimization of allosteric proteins. It applies opposing selection pressures to enhance activity and switchability of phage-encoded effectors and leverages retron-based recombineering to broadly explore fitness landscapes, introducing point mutations, insertions, and deletions. Applying this framework to the transcription factor AraC yielded near-binary optogenetic switches, with light-controlled activity spanning ~1000-fold dynamic range. Long-read sequencing across selection cycles enabled high-resolution tracking of evolving variant pools, revealing adaptive trajectories and context-dependent residue interactions. Mechanistically, we find that linker mutations promoting α-helix extension at the sensor-effector junction enhance conformational coupling between LOV2 and AraC. These variants emerge consistently across independently evolved pools, underscoring their functional relevance. Together, we develop a framework for the directed evolution of programmable allosteric switches in vivo. By coupling dynamic selection with deep mutational scanning and temporal sequencing, it enables both functional optimization and mechanistic insight into allosteric networks.

Proteins are the principal engines of cellular function, executing dynamic tasks such as catalysis, signaling, and structural organization. A defining feature of many proteins is allostery, i.e., the ability to transmit local conformational changes across distant sites to modulate protein activity in response to environmental or cellular cues. This regulatory principle, termed the "second secret of life" by Monod[1,2], underpins diverse biological processes, from gene regulation to metabolism and sensory perception, and serves as a fundamental mechanism for information processing and signal transmission in cells.

Technologies based on synthetic allosteric proteins seek to harness this regulatory mechanism for remote control of biological functions[3–7]. Optogenetics, for instance, repurposes naturally light-sensitive proteins to regulate processes such as neuronal action potential firing with precise spatial and temporal resolution[8]. The field also developed modular strategies to render non-switchable effectors light-sensitive, e.g., by inserting photosensory domains - such as light-oxygen-voltage (LOV) domains - into allosteric surface sites[6]. While the transformative potential of "synthetic allostery" is widely appreciated, engineering allosteric proteins is hampered by the difficulty of reliably

[1]Faculty of Engineering Sciences, Institute of Pharmacy and Molecular Biotechnology (IPMB), Heidelberg University, Heidelberg, Germany. [2]Zuse School ELIZA, Darmstadt, Germany. ✉e-mail: jan.mathony@uni-heidelberg.de; dominik.niopek@uni-heidelberg.de

coupling input-dependent structural changes from sensory domains across complex conformational networks in the inserted effector protein. As result, engineered receptor-effector chimeras commonly suffer from both a low dynamic range of trigger-dependent control as well as compromised overall activity[3]. Advances in molecular modeling and machine learning have improved predictions of permissive sensor-effector fusion sites[9] but are agnostic to the inserted domain and do not account for the energetic couplings or conformational dynamics

that govern allosteric switching. These limitations highlight a core challenge of synthetic allostery: its governing principles are often emergent, encoded not in isolated residues but in distributed, non-linear interactions.

Nature navigates these complex and emergent activities through variation (mutation, insertion and deletion) and selection, gradually tuning allosteric behavior in ways that support evolutionary success under shifting conditions (Fig. 1A). A canonical example is the circadian

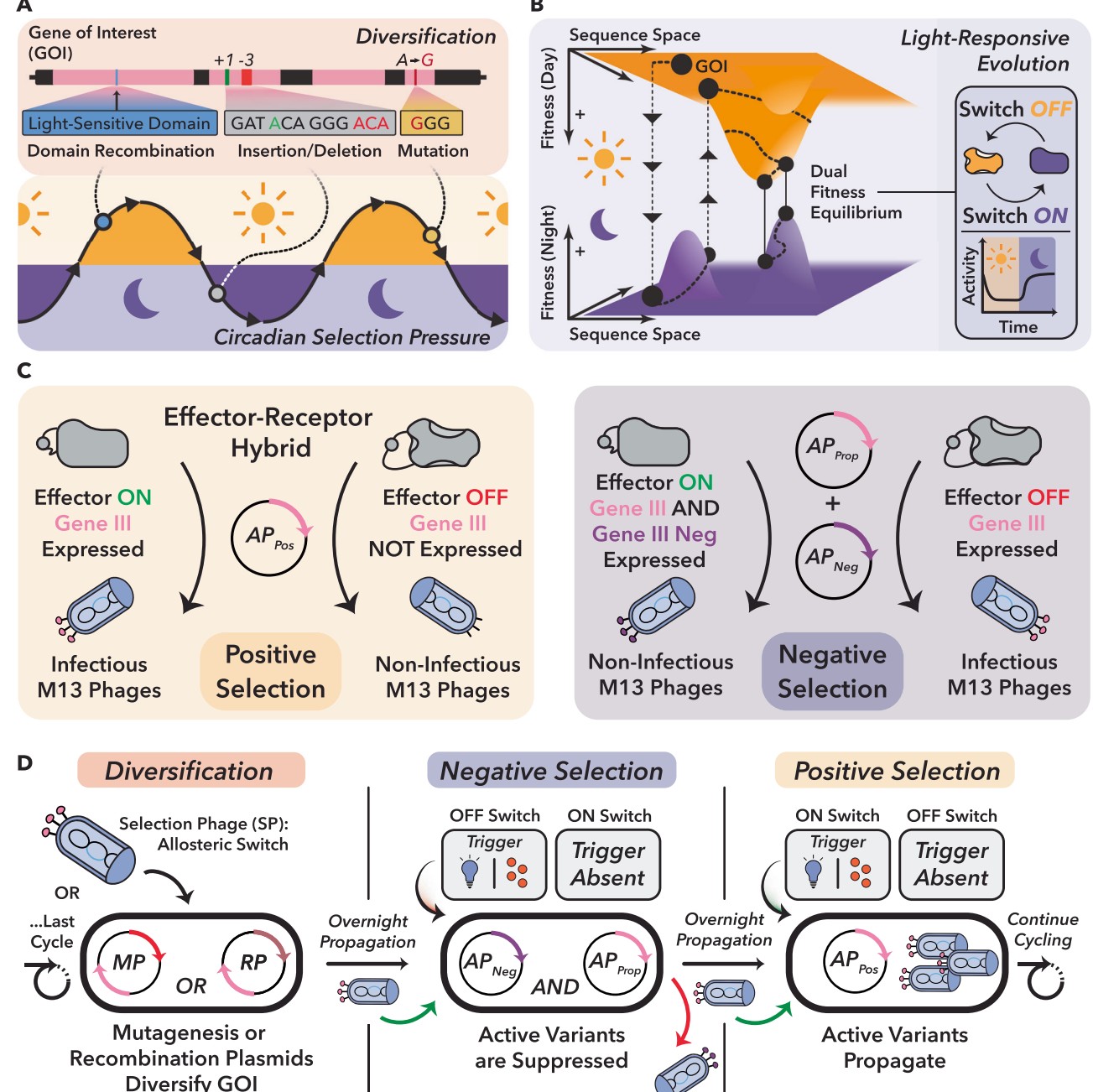

**Fig. 1 | Conceptual framework and workflow for the phage-assisted evolution of stimulus-responsive allosteric protein switches with POGO-PANCE.**
**A** Schematic of natural evolution of light-responsive proteins through domain recombination and mutation under alternating selective pressures such as day–night cycles. Such environmental cues drive the emergence of allosteric regulation by favoring variants with input-responsive activity. **B** Circadian selection imposes alternating fitness constraints on a dual-state protein, shaping evolution across fluctuating environmental conditions. **C** Engineered phage-assisted selection system towards dual-state regulation. A selection phage (SP) encodes an

allosteric switch that regulates *gIII* expression. In the positive selection circuit, switch activation induces *gIII* expression, enabling phage propagation. In the negative selection circuit, inappropriate activation triggers expression of a dominant-negative *gIII* (*gIII-Neg*) variant, suppressing phage infectivity. **D** Overview of staged POGO-PANCE regime: a selection phage (SP) library is diversified, then iteratively subjected to alternating negative and positive selection under opposing environmental states, enriching for protein variants with precise, input-dependent regulation.

clock, which evolved independently across life forms to track the day–night cycle[10], often through light-dependent allostery (Fig. 1A, B). Similar behavior emerged for various other signals, including ligands[11,12], redox state[13], or mechanical force[14]. In such systems, evolution effectively see-saws proteins across distinct fitness landscapes, enriching mutations that optimize function under alternating conditions (Fig. 1B).

We therefore hypothesized that directed evolution under dynamic selection regimes could reveal the complex fitness landscapes of allosteric protein behavior and thus uncover context-dependent sequence-function relationships beyond the reach of rational design[15,16]. To test this, we employed phage-assisted evolution, a powerful and customizable platform for in vivo protein optimization[16,17]. By coupling effector protein function to M13 phage propagation, this approach enables diversification and selection directly in *Escherichia coli* (*E. coli*), providing a highly effective system for protein engineering[18–21]. In these systems, effector activity is typically coupled to phage fitness by placing *Gene III* (*gIII*), an essential infectivity factor of M13 phage encoding protein III (pIII), under the control of the evolving protein. Effector variants that induce *gIII* expression enable productive phage propagation, thereby forming the basis of selection. Existing phage-assisted evolution systems, however, focus on optimizing a selected activity under relatively static conditions, rendering them ill-suited for evolving switchable proteins.

Here we present POGO-PANCE (Protein-switch On/Off Gene Optimization using Phage-Assisted Non-Continuous Evolution (PANCE)), a directed evolution strategy that mimics dynamic selection regimes such as the day–night cycles that gave rise to circadian clocks (Fig. 1A). To achieve this, POGO-PANCE alternates between temporally separated, mutually exclusive positive and negative selection steps, aligned with the presence or absence of an input signal (Fig. 1B–D). In positive selection, effector activity induces expression of *gIII*, thereby enabling propagation[16]. In negative selection, the same logic is inverted: effector activity drives a dominant-negative *gIII* variant (*gIII-Neg*) that blocks phage infectivity[22], while wild-type *gIII* is co-supplied to permit propagation of inactive variants. Although prior PACE and PANCE efforts have also employed alternating positive and negative selection, these approaches were largely designed to reconfigure intrinsic protein properties, such as reducing promiscuous enzymatic activities[23] or altering ligand binding[24], rather than to evolve proteins that respond to an external cue. By toggling the input signal across these steps and applying iterative mutagenesis, POGO-PANCE co-navigates two alternating fitness landscapes, promoting the emergence of effector protein variants that are both highly active and input-responsive.

A critical determinant of directed evolution success is the diversity of the genetic pool subjected to selection. In standard phage-assisted evolution systems, mutagenesis plasmids (MPs) increase point mutation rates by perturbing DNA replication and repair[25]. However, they introduce strong mutational biases and cannot generate intentional, in-frame insertions or deletions (InDels), significantly limiting the accessible protein sequence space. We therefore developed RAMPhaGE (Recombitron-Assisted Multiplexed Phage Gene Evolution), a retron-based in vivo diversification method capable of introducing point mutations and InDels at tunable frequencies, thus unlocking regions of the protein sequence space previously inaccessible to phage assisted evolution.

To demonstrate these concepts, in this work we apply POGO-PANCE and RAMPhaGE to the arabinose-responsive transcription factor AraC, a well-characterized bacterial effector protein. Through iterative evolution cycles, we obtain light-responsive AraC variants exhibiting near binary switch-like behavior and dynamic control spanning over three orders of magnitude. Tracking evolving phage pools via short- and long-read sequencing reveals key adaptive trajectories and pinpoints structural features and epistatic interactions

that underlie effective allosteric switching. Together, POGO-PANCE and RAMPhaGE not only enable evolution of programmable molecular switches, but also open a window into the emergence and architecture of allosteric networks.

## Results
### Evolution of highly active AraC variants as evolutionary stepping stones

To experimentally evaluate POGO-PANCE, we sought to adapt a well-characterized genetic switch, the *E. coli* transcription factor AraC, for optogenetic control. AraC regulates pBAD promoter activity by switching from a repressive to an activating state in response to L-arabinose (Supplementary Fig. 1A) and has been widely used in microbiology for ligand-inducible gene expression[26]. Native AraC functions as a dimer in both repressing and activating states[26]: without L-arabinose, it favors $O_2$–$I_1$ half site occupancy in the pBAD promoter to repress expression, while arabinose binding induces a conformational change that shifts occupancy to the $I_1$–$I_2$ half sites to release the loop (Supplementary Fig. 1A). In this activating configuration, AraC directly engages the RNA polymerase α-CTD to recruit and stabilize the RNAP, driving transcription initiation downstream. Importantly, AraC has been shown to be highly evolvable, capable of adapting to novel inducers[27,28], thus making it an ideal candidate for input reprogramming. Light, as an input, offers superior spatiotemporal precision compared to chemical inducers and integrates readily into the batch culture format of PANCE. We therefore applied POGO-PANCE to engineer potent optogenetic AraC variants, both to explore the feasibility of this concept and to generate a tool for tight, light-dependent gene expression control.

To enable solely light-based control, however, we first needed to eliminate AraC's native dependence on arabinose and rewire its regulatory logic for non-native input. To this end, we designed a PANCE circuit that separates evolution into distinct mutagenesis and selection phases. For mutagenesis, we adapted the previously published drift plasmid 6 (DP6)[25] for IPTG-inducible mutagenesis[29] and anhydrotetracycline-dependent *gIII* expression (Fig. 2A). For positive selection, we constructed an Accessory Plasmid (AP) encoding *gIII* under the control of the pBAD promoter and confirmed that it enables propagation of active AraC variant-encoding phages, supporting approximately 20-fold higher propagation in the presence versus absence of arabinose (Fig. 2B and Supplementary Fig. 1B).

Next, to evolve constitutive AraC, we applied two distinct selection regimes (R) across three independent evolution pools each to promote diverse adaptive trajectories. In one condition set, phages were selected entirely in the absence of arabinose (R1-R3), while in the other, arabinose levels were incrementally reduced over five passages (100 μM to 0 μM, R4-R6) (Fig. 2C). To maintain selective pressure without population saturation, we imposed sequential bottlenecks between mutagenesis and selection by controlling input phage levels (Supplementary Note 1).

Over five rounds of mutagenesis and selection, phage output titers remained high (Supplementary Fig. 1C), with progressively increased propagation observed after each round (Supplementary Fig. 1D). This suggested the emergence of strong constitutive activity. Phage pool sequencing, i.e., Sanger-sequencing of the AraC PCR amplicons derived from the evolved phage pool, indicated diverse evolutionary lineages (Fig. 2D). We selected one representative variant from each of the six pools (R1-R6) for further testing using a plasmid-based reporter assay in *E. coli*, in which AraC activates a pBAD-driven red fluorescent protein (Fig. 2D and Supplementary Fig. 1E). Although overnight cultures lacked arabinose or IPTG (which controls AraC expression via a LacI-regulated promoter), they nonetheless appeared bright red for five of the six variants (Fig. 2E). This indicated that these evolved AraC proteins exhibit high constitutive activity even at low

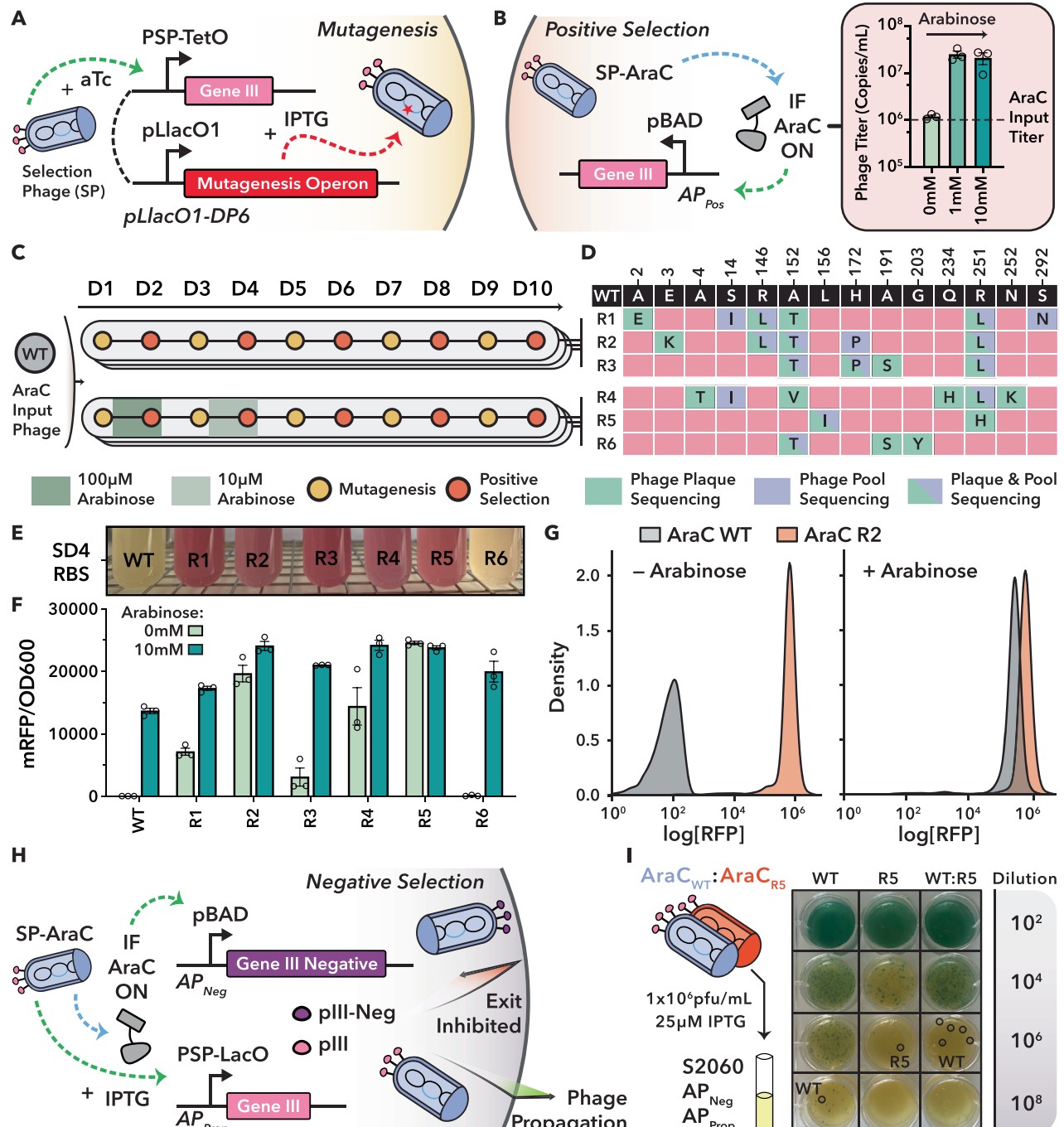

**Fig. 2 | Phage-assisted non-continuous evolution of hyperactive AraC variants.**
**A** Schematic of the mutagenesis system. Infection with a selection phage (SP) and addition of anhydrotetracycline (aTc) induces *gIII* expression, while IPTG induces a mutagenesis operon. **B** Positive selection circuit. SPs encoding active AraC variants induce *gIII* expression via the pBAD promoter. Right: Arabinose-dependent propagation of an SP encoding WT AraC, demonstrating induction-dependent propagation. Bars represent mean ± Standard Error of the Mean (S.E.M.) of three biological replicates, as quantified by qPCR. Corresponding phage plaque assay shown in Supplementary Fig. 1B. **C** Summary of the AraC PANCE campaign. Phage titers from each day (D1-10) are shown in Supplementary Fig. 1C, D. **D** Mutations in evolved AraC variants. Pooled phage populations and one individual plaque from the final selection day (D10) were sequenced. **E** WT AraC and evolved variants (R1−R6) were expressed from a LacI-dependent expression cassette containing the SD4 RBS and co-transformed with a pBAD regulated mRFP reporter plasmid (Supplementary Fig. 1E). Cultures were grown without inducers **F** Quantification of

AraC variant activity. AraC variants were expressed as in (**E**), but using the weaker sd2 RBS. RFP and $OD_{600}$ were measured in a plate reader. Data shown as mean ± S.E.M. from $n = 3$ biological replicates. **G** Flow cytometry analysis of WT AraC and R2 variant. Reporter fluorescence was quantified in presence or absence of 16 mM L-arabinose. Gating strategy described in Supplementary Fig. 2. **H** Schematic of the negative selection circuit. SPs encoding active AraC variants induce expression of *gIII-Neg* to produce Protein III Neg (pIII-Neg) from the pBAD promoter, prohibiting phage exit. Concurrently, IPTG induction drives WT *gIII*. **I** Head-to-head competition assay between WT AraC and R5 under negative selection. Host cells harboring $AP_{Prop}$ and $AP_{Neg}$ were infected with (i) $1 \times 10^6$ pfu/mL of either phage individually or (ii) with $5 \times 10^5$ pfu/mL of each phage combined. Cultures were supplemented with 25 μM IPTG (0 mM Arabinose) at the time of phage infection, followed by plaque assay. Resulting plaques were sequenced (circled in black) to determine dominant variants. Expanded results in Supplementary Fig. 3B, with sequence confirmation in Supplementary Fig. 3C.

expression levels driven solely by leaky transcription from the LacI repressed promoter.

To facilitate quantitative comparisons between evolved variants, we introduced a substantially weaker ribosome binding site (RBS) upstream of AraC[30]. Under these conditions, all six evolved variants induced higher reporter expression levels than wild type AraC (WT) in the presence of arabinose, and five of six showed constitutive activity in its absence (Fig. 2E, F). R2 and R5 even surpassed the output of fully induced WT, suggesting that accumulated mutations conferred potent, ligand-independent activation (Fig. 2F, G; Supplementary Fig. 2). Both variants carried substitutions previously linked to constitutive activity: R2 bore A152T and R251L, which are thought to disrupt inhibitory interdomain contacts[31], while R5 combined R251H with L156I, the latter associated with altered ligand specificity[32]. Additional mutations in the auto-inhibitory N-terminal arm (a flexible extension that folds back onto the DNA-binding domain to stabilize the repressive state), such as A2E, E3K, A4T, and S14I, further support a shared mechanism of weakened repression[33]. Together, these patterns highlight diverse but convergent routes to constitutive activation.

## POGO-PANCE yields potent optogenetic switches and probes allosteric networks

Having created particularly strong, constitutive AraC variants as evolutionary stepping stones, we next assembled circuits to implement the full POGO-PANCE pipeline for evolving light-switchable AraC variants (Fig. 1C, D). We constructed an accessory plasmid (AP_Neg) in which *gIII-Neg* is expressed under control of the pBAD promoter (Fig. 2H), and then co-transformed this plasmid with AP_Prop (Accessory Plasmid for Propagation), a LacI-IPTG controlled circuit for tunable *gIII* expression, into S2060[34] (Supplementary Fig. 3A). This configuration enables tunable negative selection against AraC activity, with IPTG induction modulating basal phage propagation rates.

To validate the negative selection host cells, we ran experiments in which phages encoding WT or the evolved, hyperactive R2 or R5 variants were propagated under mild IPTG induction (25 μM) and in absence or presence of arabinose (Fig. 2H, Supplementary Fig. 3B). All three phages showed a clear arabinose-mediated propagation deficit, with the highly active R2-encoding phages exhibiting almost complete propagation inhibition even in the absence of arabinose (Supplementary Fig. 3B). Mixing WT encoding phage with R2 or R5 phage in equal quantities resulted in arabinose-dependent propagation again (Supplementary Fig. 3B). Importantly, sequencing five single phage plaques from the mixed AraC WT:R5 group without arabinose revealed that all were of WT background (Fig. 2I, Supplementary Fig. 3C), confirming that the negative selection regime effectively selects against active variants in favor of inactive variants in the same pool.

To explore whether evolved AraC variants could serve as a foundation for light-switchable gene expression control, we inserted the blue light-responsive LOV2 domain from *Avena sativa* phototropin-1[6] (referred to as LOV) into each of the six variants at a previously established allosteric site immediately following residue S170[35,36], flanked by SG and GS linkers (Supplementary Fig. 4). As our aim was to use domain insertion as the primary regulatory modality for investigating allosteric regulation in AraC, we retained the native Arabinose binding domain rather than removing it[35]. Light-induced unfolding of LOV terminal α-helices is a well-established mechanism, commonly harnessed in synthetic allostery for generating local disorder and conveying an allosteric signal. In this configuration, the resulting AraC–LOV hybrids function as light-inactivated, OFF-logic switches, a mode that is well suited for scenarios where gene expression is high by default but needs to be shut off in a temporally or spatially controlled manner, for example to terminate expression at defined time points. However, existing AraC-LOV hybrids have shown limited dynamic range and compromised activity[35,36]. As initial test cases, we evaluated phage propagation for R2-LOV and R5-LOV due to their strong

constitutive activity and distinct mutational backgrounds. Both showed minimal or no propagation under light or dark conditions (Supplementary Fig. 5A–D), indicating that domain insertion disrupted AraC function and failed to confer robust photoswitching under selection conditions.

We therefore returned to assessing all six LOV-inserted variants in a plasmid-based mRFP reporter assay using the same expression conditions as in the original constructs (see Fig. 2F). Without IPTG induction, none of the AraC-LOV variants produced measurable output in either light or dark, confirming that LOV insertion severely impaired arabinose-independent transcriptional activation from the pBAD promoter (Supplementary Fig. 5E; compare to Fig. 2F). However, high-level expression with 400 μM IPTG restored dark-state activity in four LOV hybrid variants (R1, R2, R4, and R5), each showing reduced reporter levels under blue light, consistent with residual light-dependent regulation (Supplementary Fig. 5F). This outcome is prototypic for the engineering of allosteric effectors, where sensory domain insertion often severely reduces effector function and residual activity can only be detected upon strong protein overexpression[3]. Although this functional rescue upon overexpression does not correct the underlying defect, it exposes if a chimera could serve as a starting point for directed evolution. Because R2- and R5-LOV exhibited noticeable switchability upon strong overexpression and enabled phage propagation on our selection circuits, we hypothesized that these would be ideal candidates for optimization by POGO-PANCE.

We next evolved both variants using three distinct selection strategies, each run in duplicate pools (Fig. 3A and Supplementary Fig. 6). The first POGO-PANCE regime (P1) consisted of three cycles of the following steps: (1) mutagenesis, (2) negative selection under blue light to eliminate leaky or constitutive variants, enriching those with low activity in the presence of the trigger, and (3) positive selection to favor variants that exhibit potent activity in the dark, thus promoting switch-like behavior. The second regime (P2) began with two cycles of (1) mutagenesis and (2) positive selection to maximize activity, followed by two cycles of (1) mutagenesis, (2) negative selection in presence of light and (3) positive selection in absence of light. The third regime (P3) was identical to the first, but reversed the light conditions during selection and served as a control with non-aligned input. Between steps, phages were diluted ~1:100, to preserve gene pool diversity (Supplementary Note 1). Of note, a recombination-driven excision of the LOV domain was observed for R5-LOV P2.1, while plaque assays revealed washout-like propagation failure for both R2-LOV P3.2 and R5-LOV P3.1, eliminating these three pools from further consideration (see "Discussion").

Upon completion of the evolution regimes, we analyzed the endpoint mutational landscape across the nine remaining phage pools (see "Methods"; LOV residue numbering corresponds to full-length phototropin-1). Mutations were broadly distributed across the AraC-LOV fusion proteins (Supplementary Fig. 7), suggesting that both domains co-evolved under selection. In the AraC section, several residues emerged as recurrent hotspots, including E144 and R146 positioned near the arabinose-binding pocket and directly involved in ligand recognition, as well as V223 and S225, located within the C-terminal DNA-binding domain's second helix-turn-helix motif[26]. These latter residues lie adjacent to a dynamic region implicated in DNA binding and transcriptional regulation[37]. In the LOV domain[38–43], recurrent mutations were found at D501, E541, and A543 as well as in the right AraC-LOV junction (Serine in right linker, position 2; SRL2) (Fig. 3B). D501 sits near the FMN chromophore and participates in the hydrogen-bonding network that stabilizes the chromophore binding pocket[38], while E541, A543 and SRL2 lie in (or close to) the C-terminal Jα helix, a structural element that undocks from the LOV core upon light activation[40]. Thus, these mutations lie in highly dynamic regions and apparently comprise a critical interface that couples LOV photodynamics to AraC allostery. Notably, except for

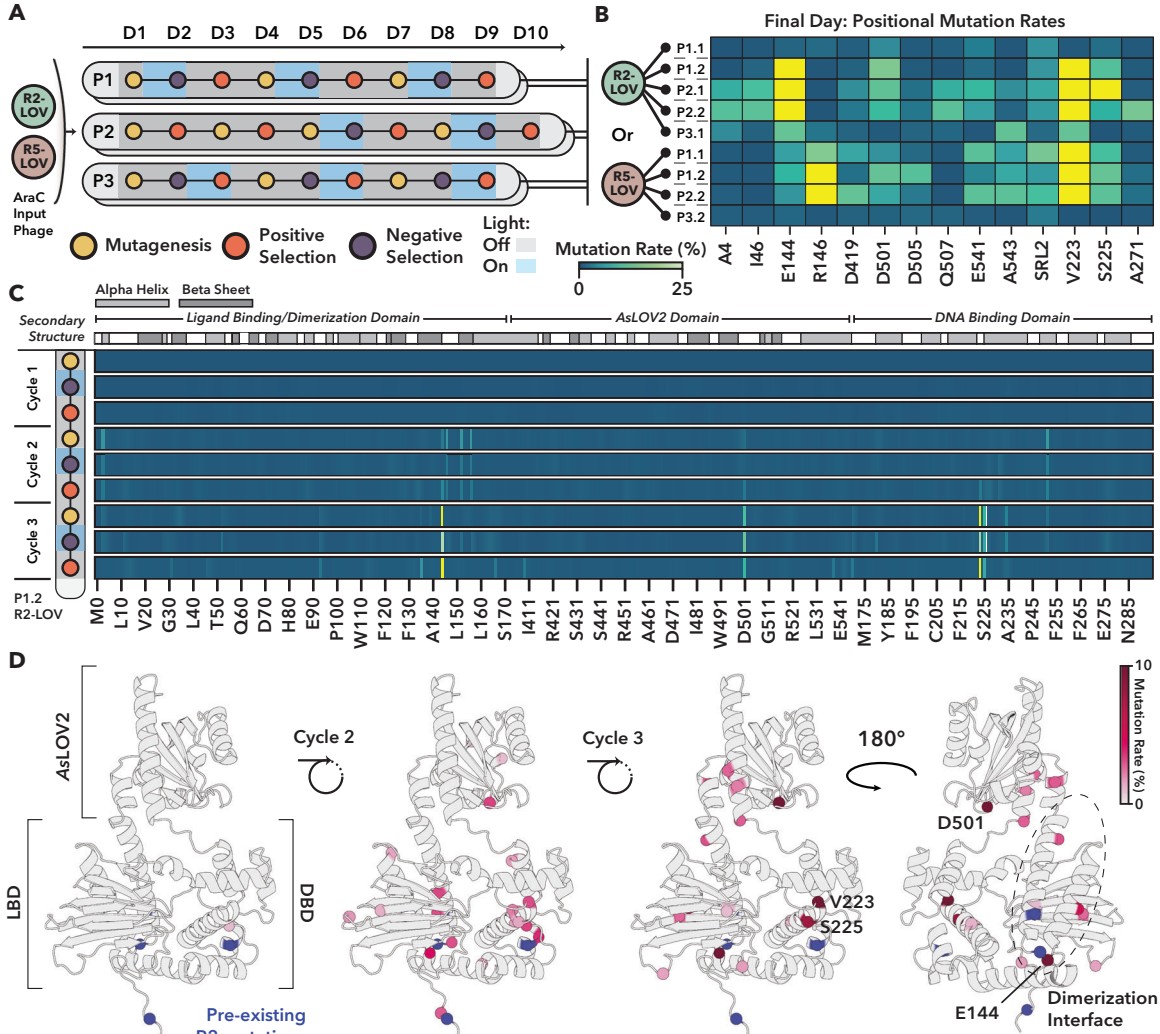

**Fig. 3 | Directed evolution of light-sensitive AraC S170 AsLOV2 variants using POGO-PANCE. A** Summary of AraC-LOV POGO-PANCE campaigns. Mutagenized phage libraries encoding AraC-R2-LOV or AraC-R5-LOV were subjected to iterative rounds of positive and negative selection under alternating light and dark conditions (for variants see Supplementary Fig. 5E, F). Three selection strategies were tested, each performed in duplicate pools. The illumination set-up and phage titers are shown in Supplementary Figs. 5B and 6A, B, respectively. **B** Endpoint positional enrichment analysis of mutations across evolved variants quantified by Nanopore sequencing. Mutation rates at each residue within AraC-LOV are displayed as the percentage of total reads per position, with residues exceeding 25% shaded in yellow. Positions that have an enrichment >10% in at least one of the pools are shown, excluding position R38 which arose as a systematic artifact in Nanopore sequencing (Supplementary Fig. 20). **C** Temporal mutation enrichment during the R2-LOV P1.2 evolution campaign quantified by Nanopore sequencing. Mutation rates at each residue within AraC-LOV are displayed as the percentage of total reads per position, relative to enrichment observed after mutagenesis round 1 (see Supplementary Fig. 20). Residues exceeding 25% are shaded in yellow. Secondary structure annotations for AraC-LOV are shown above. **B**, **C** The shown "Mutation rate (%)" label applies to both panels. **D** Mutation rates observed after positive selection steps for the R2-LOV P1.2 evolution in **c** are mapped onto the Alphafold3 predicted structure of AraC-LOV (Supplementary Fig. 4). Spheres at the alpha-carbon refer to residues enriched above 1% and are colored according to their enrichment; residues exceeding 10% are additionally shaded in burgundy and are annotated. The AraC Ligand-Binding Domain (LBD), AraC DNA Binding Domain (DBD), and AsLOV2 domain are indicated with brackets, and the native AraC dimerization interface is circled with a dashed line. Pre-existing mutations in R2 (see Fig. 2) are marked in blue.

A4 and R146 in R5-LOV, which was associated with increased activity in the R2 variant (Figs. 2D and 3B), none of these mutations were present in any prior PANCE pool during selection for constitutive mutations. This mutational divergence suggests the emergence of distinct, context-dependent mechanisms, potentially enhancing domain insertion compatibility or enabling light-dependent switching behavior.

To understand how these patterns emerged and how selection shaped the switching phenotype, we performed long-read sequencing[44] across all mutagenesis and selection steps of one evolution pool (P1.2, R2-LOV variant; Fig. 3C). Distinct positional enrichments were already apparent by the second cycle, but most recurrent variants reached strong enrichment only after three full rounds,

underscoring the cumulative effect of sequential selection. Several positions showed transient enrichment or depletion across different steps, indicating that both negative and positive selection cycles actively sculpted sequence space. When mapped onto an AlphaFold3-inferred structural model of the wild-type AraC S170−LOV structure, mutations formed a diffuse pattern spanning both domains and involving α-helices, β-strands, and linker regions (Fig. 3D, Supplementary Figs. 4 and 7). These observations support the emergence of a distributed allosteric network, in which light-driven conformational changes in LOV are transmitted through a structurally integrated interface to modulate AraC activity.

Sanger sequencing of 36 AraC-LOV variants isolated from the phage pools revealed a total of 28 genetically unique variants (Fig. 4A),

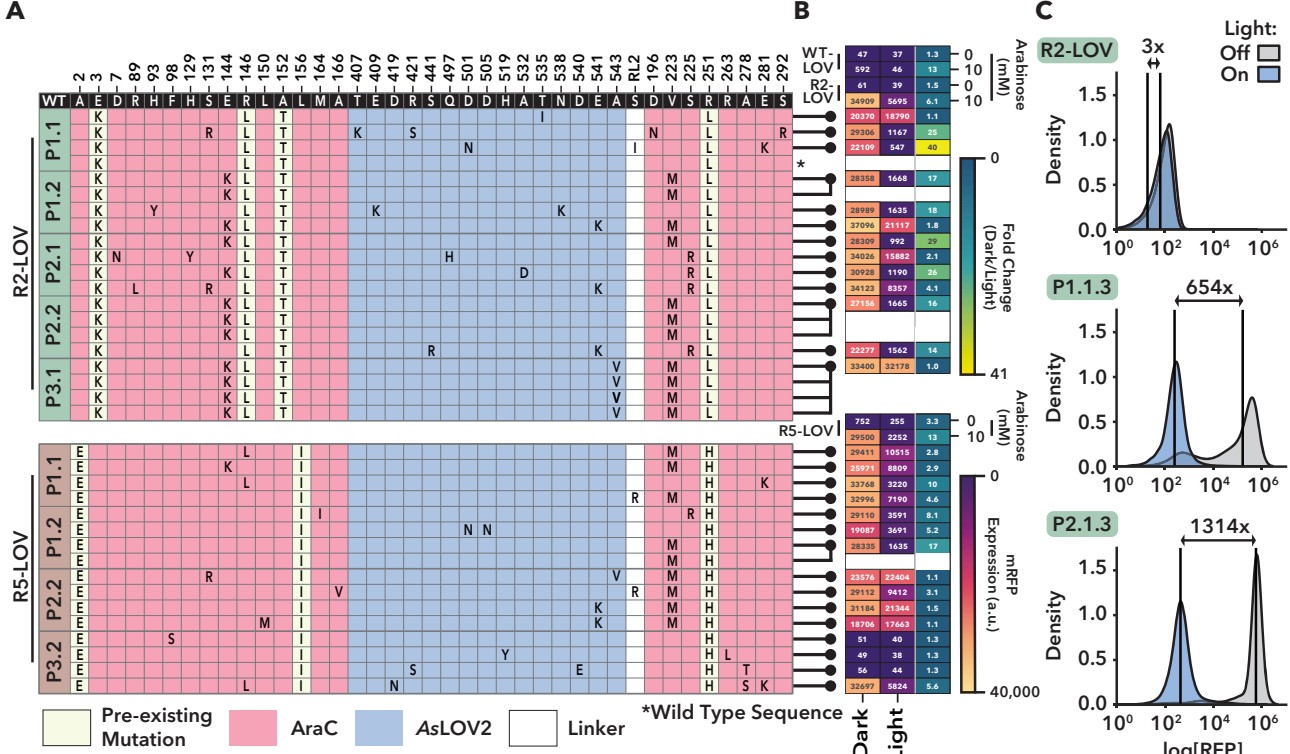

**Fig. 4 | Functional characterization of evolved AraC-LOV variants derived from POGO-PANCE. A** Genotypes of individual endpoint variants. Four phage plaques from the final day of each evolution pool were isolated, cloned into an expression vector, and sequenced. Mutations across the LOV-flanking linker, LOV domain, and AraC regions are indicated. **B** Heatmap of RFP reporter activity under light and dark conditions. Variants (linked to identities in **A**) were expressed in *E. coli* assessed for activity using the RFP reporter assay. Cultures were incubated in the light or dark for 18 h, followed by measurement of mRFP fluorescence and $OD_{600}$ in a plate reader. Variants were incubated with no Arabinose or IPTG added, and side-by-side compared to WT, R2-, and R5-LOV controls with or without L-Arabinose. Mean fluorescence values, normalized to $OD_{600}$, were obtained from the average of three biological replicates (first two columns); fold change values (dark/light) are shown in the third column. All values are displayed within their respective boxes. **C** Flow cytometry analysis of light-dependent RFP expression in top-performing variants. Cultures were prepared as described in (**B**) without arabinose or IPTG, for the R2-LOV, R2-LOV P1.1.3, and R2-LOV P2.1.3 variants to assess switching behavior based on mRFP reporter fluorescence. Median mRFP fold changes between dark and light conditions are shown.

with mutational patterns aligning well with the long-read sequencing data (compare Figs. 3B, 4A, and Supplementary Fig 8). Several mutations showed reproducibly high convergence on a few residues and exhibited repeated mutational co-occurrence patterns in the AraC protein part, despite these residues being distant in the AraC structure (Fig. 4A, Supplementary Fig. 9A, B). For example, E144K frequently appeared together with V223M in R2 backgrounds but was rarely detected in R5 (Fig. 4A; Supplementary Fig. 9B). Conversely, R146L and V223M commonly co-appeared in R5 pools yet were also recovered independently (Fig. 4A; Supplementary Fig. 9B). V223M arose across all three evolution regimes (P1–P3), whereas combinations such as V223M and S225R were rarely observed, suggesting mutually exclusive fitness trajectories (Fig. 4A; Supplementary Fig. 9B).

Subsequent plasmid-based mRFP reporter assays confirmed that all R2-LOV and R5-LOV variants derived from the aligned P1 and P2 regimes exhibited substantially elevated dark-state activity compared to their non-evolved counterparts (Fig. 4B). Importantly, light-dependent repression was broadly preserved and, in many cases, the magnitude of light responsiveness increased, with all but one R2-derived variant exhibiting higher dark/light fold-changes than their parental construct and several R5-derived variants showing similarly robust switching behavior (Fig. 4B). Closer investigation of two particularly promising R2-LOV variants, namely clone 3 from the P1.1 and P2.1 pools, by flow cytometry confirmed high dark-state activity and exceptionally strong photoswitching, with median fluorescence shifting by about three orders of magnitude upon illumination (Fig. 4C). This indicated that the mutations that had emerged during POGO-

PANCE substantially improved allosteric switching. In contrast, the control evolution condition (P3) predominantly yielded non-switching variants, underscoring the critical importance of aligning the light input with the appropriate selection step (Fig. 4A, B). Notably, the evolved AraC–LOV variants function as blue-light-inactivated switches, with high dark-state activity that is strongly reduced upon illumination. For contextual reference, a comparison under matched assay conditions with a previously reported, light-activated VVD–AraC "BLADE" protein[35] (Supplementary Fig. 10A, B) was performed. Compared to our lead variant R2-1.1.3, BLADE exhibited both a considerably lower activity in its ON-state and lower dynamic range of light control (Supplementary Fig. 10B), indicating that the evolved AraC–LOV switches provide substantially enhanced absolute output and light-dependent modulation relative to BLADE.

Collectively, these data indicate that POGO-PANCE is a highly effective approach for evolving optogenetic proteins with exceptional performance. Beyond allosteric switch engineering, POGO-PANCE also provides a powerful lens to observe and interrogate allosteric networks by evolving them in real time, revealing how effective communication and signal relay emerges across protein domains from a network of seemingly unrelated residues.

## RAMPhaGE: A highly effective, targeted gene diversification strategy for PANCE

MPs such as MP6 and DP6 are widely used to diversify transgenes and generate gene pools for phage-assisted evolution, but come along with significant limitations that constrain access to broad protein

sequence landscapes. First and foremost, MP/DP systems are restricted to point mutations and cannot introduce intentional, in-frame insertions or deletions (InDels), which are particularly relevant for engineering allosteric switches where tuning the length, orientation, or spacing between communicating structural elements is often critical[45]. Second, the mutations generated are highly biased toward specific base changes[25], thus sampling only a narrow portion of the sequence landscape. Finally, although phage libraries generated by direct cloning generally enable broader diversification, their composition and complexity are restricted to what is achievable in a single diversification step, i.e., different variants cannot be layered without re-cloning the input phage[46].

To overcome these limitations, we developed a versatile in vivo transgene diversification strategy based on retron-mediated recombineering[47], a genome editing method capable of introducing targeted substitutions, insertions, and deletions[48] to enable iterative, layered diversification through propagation on retron-carrying hosts. Retrons encode an RNA template and a reverse transcriptase that together produce intracellular single-stranded DNA (ssDNA) from a plasmid, providing a recombination donor during replication[47]. To adapt retrons into a targeted in vivo diversification strategy of M13 phage carried transgenes, we used a Recombitron: a fully genetically encoded operon combining the *Eco1* retron, single-stranded annealing protein (CspRecT), and a dominant-negative variant of the *E. coli* mismatch repair suppression protein *MutL* (Fig. 5A). Although previously validated in double-stranded DNA phages, recombitrons have not yet been shown to function in ssDNA phages such as M13[47].

We therefore engineered an IPTG-inducible Recombitron carrying a 90-nucleotide (nt) retron template encoding a silent single-nt substitution in the AraC-LOV transgene. The Recombitron was expressed in *E. coli* S2060 and M13 phages were propagated overnight on these hosts (Fig. 5A). Pooled Sanger sequencing revealed precise, 100% editing of the target site (Fig. 5B and Supplementary Fig. 11A, B). No editing was observed in controls carrying either (i) a non-targeting retron, (ii) the Recombitron with the reverse transcriptase (RT) disabled by mutation or (iii) a functional Recombitron with a retron encoding the WT variant (Fig. 5B and Supplementary Fig. 11A, B), confirming that editing was dependent on the presence of all key Recombitron components. Notably, phage titers in the targeting retron group remained at input levels (Supplementary Fig. 11a), suggesting that Recombitron induction can transiently constrain propagation. However, when we repeated the experiment using a higher input titer of $10^{10}$ pfu/mL, long-read sequencing revealed >70% editing efficiency at the target site (Supplementary Fig. 11C), indicating that this limitation can be readily overcome by adjusting input phage dose.

Next, we adapted the Recombitron system for the generation of complex M13 mutational libraries. To facilitate library design, we developed a Golden Gate-compatible recombineering plasmid (RP) featuring a *ccdB* dropout cassette and a custom software tool that automates the generation of barcoded retron templates, enabling direct oligo pool synthesis with precise control over mutation identity and distribution across M13-encoded transgenes (Supplementary Fig. 12 and "Methods"). Using this framework (Fig. 5C), we constructed two specialized libraries. The first, a deep mutational scanning (DMS) library, encoded all (2820) possible single amino acid substitutions across the LOV domain in AraC-LOV. The second library mediated insertions, deletions, and substitutions selectively at the sensor–effector junctions of the AraC–LOV fusions ("linker library"), thus altering the connecting SG and GS linkers and/or linker-adjacent residues on either side of the LOV insert. Specifically, the library included small deletions of one to three amino acids at the junctions, as well as a diverse panel of substituted or inserted linkers that varied in length and composition (from no linker at all to six inserted

residues). These comprised single-residue linkers with random amino acids, glycine–serine–based flexible linkers, proline-rich rigid linkers, and also incorporated LOV variants with truncated or extended Jα helices. Supplementary Data 6 summarizes the variants encoded in the linker library.

To assess editing efficiency and mutational outcomes, we propagated AraC-LOV-encoding phages overnight on the DMS library hosts and pLlacO1-DP6 cells followed by short- and long-read deep sequencing. DMS library editing produced the programmed mutations with high frequency (>40% of reads; Fig. 5D). Further analysis revealed that the average mutation rate per retron-targeted position was -0.4% —more than triple observed for pLlacO1-DP6 (Supplementary Fig. 13A).

In contrast to the highly biased mutational profile generated by pLlacO1-DP6, the DMS library produced broad and relatively uniform coverage across the full spectrum of encoded amino acid and codon substitutions (Fig. 5E, Supplementary Fig. 13B, C). Variant frequencies in the DMS phage library closely matched those in the retron-encoding plasmid library (Supplementary Fig. 13D; Spearman $r = 0.77$; Fig. 5F, top). Because codon substitutions require between one and three mutations, single-base codon edits occurred at approximately twice the rate of double or triple substitutions, consistent with known recombination biases favoring smaller edits[47] (Supplementary Fig. 13E). Off-target mutation rates outside the retron-targeted LOV region were minimal, as confirmed by short-read sequencing at the AraC-LOV junction and long-read analysis of the full transgene (Fig. 5F and Supplementary Fig. 14).

Excitingly, the linker library introduced not only a diverse spectrum of amino acid substitutions but also a broad range of insertions and deletions that span up to five amino acids—demonstrating that even 15 bp changes within the 90 nt retron payload could be efficiently installed (Supplementary Fig. 15A). Variant frequencies in plasmid input and phage output libraries were again correlated (Supplementary Fig. 15B; Spearman $r$ of 0.79 and 0.41 for the left and right AraC-LOV junction), and editing rates depended on the encoded InDel size as expected (Supplementary Fig. 15A). Interestingly, after a single round of overnight propagation on Recombitron-expressing hosts, mutations were typically confined to one junction (either left or right), while double-mutants were rare (Supplementary Fig. 15C). This is consistent with the biology of M13 infection[49]; M13 enforces strong superinfection exclusion, which prevents additional phages from entering an already infected cell, and at the high multiplicity of infection (MOI) used here most host cells become infected immediately. Consequently, phages at this MOI cannot move between hosts or accumulate multiple independent mutations within a single round, and the appearance of double-mutant phage is correspondingly rare.

That said, retron libraries enable sequential, layered mutagenesis on demand. To demonstrate this, we first introduced DMS edits within the LOV coding region by overnight phage propagation on host cells expressing the DMS retron library. The resulting phage progeny were then transferred to hosts carrying the linker library to diversify the AraC-LOV junctions. This two-step protocol yielded cumulative editing, with widespread point mutations across LOV and substantial linker diversification at the insertion boundaries after propagation on both host libraries (Supplementary Fig. 15D and Fig. 5D).

We refer to this strategy as RAMPhaGE (Recombitron-Assisted Multiplexed Phage Gene Evolution), an M13-based transgene diversification platform that merges the precision of retron-guided editing with the scalability of in vivo phage-assisted evolution.

## Integrating RAMPhaGE and POGO-PANCE to evolve sensor-effector junctions for enhanced allosteric coupling

Sensor-effector junctions are key determinants of allosteric communication and switch fidelity in chimeric proteins[50], and were

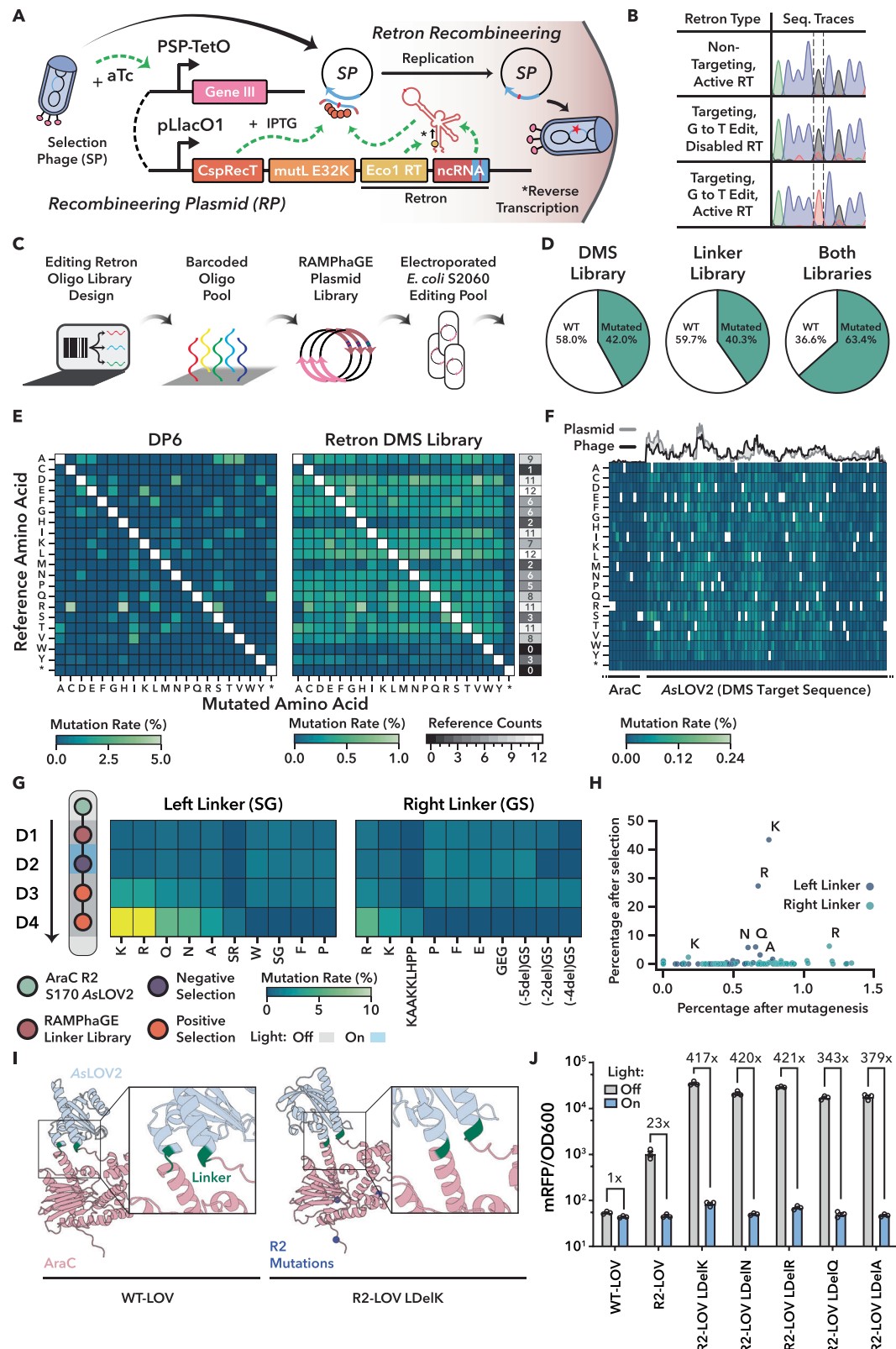

consistently targeted in all backgrounds during POGO-PANCE evolution (Figs. 3 and 4). Building on this observation, we integrated RAMPhaGE into the POGO-PANCE framework for targeted evolution of linker regions flanking the LOV insertion in AraC. As a starting point, we returned to the unoptimized R2-LOV variant (Fig. 3A and Supplementary Fig. 5E, F) and generated a diverse phage population encoding systematic modifications to the LOV-flanking linkers with the retron

linker library. These included flexible glycine-serine motifs, rigid proline-rich sequences, and all possible single-residue substitutions as confirmed by deep sequencing (Supplementary Fig. 15A, B).

Next, the resulting phage library was split into two replicate pools and subjected to one round of negative selection under blue light to remove leaky variants, followed by two rounds of positive selection in the dark to enrich for constructs with high activity in absence of the

**Fig. 5 | Directed evolution of AraC-LOV variants using RAMPhaGE and POGO-PANCE. A** Overview of the retron recombineering system. **B** Sanger sequencing chromatograms of phage pools analyzed after overnight editing with 90 bp retrons under the indicated conditions. The G to T edit position is indicated. A non-targeting retron control without homology to the transgene is included. **C** Schematic of RAMPhaGE editing library construction. **D** Percentage of edited versus WT phages from retron libraries targeting either the LOV domain (DMS Library), the linker regions, or both. **E** Comparison of amino acid-level mutation coverage within the LOV domain following a single passage using either the pLlacO1-DP6 mutagenesis system or the retron-encoded DMS library. The gray heatmap indicates the occurrences of each amino acid in the LOV reference. **F** Amino acid-level mutation rates per position after a single phage passage on the DMS library carrying hosts, for a region around the left AraC-LOV boundary. The top line plot compares the relative nucleotide mutation frequency encoded in the underlying plasmid DMS to those observed in the edited phage population, with both distributions normalized to sum to 1. **G** Enrichment analysis from a POGO-PANCE/RAMPhaGE evolution campaign using R2-LOV as input. Mutation rates observed in the left and right linkers of AraC-LOV were quantified by Nanopore sequencing. The top 10 linkers with the highest variance across all POGO-PANCE steps are shown for each site, with residues exceeding 10% shaded in yellow. The SG variant of the left linker refers to a silent mutation. **H** Scatter plot comparing variant frequencies after mutagenesis to the final positive selection. Variants observed in the left and right linker are indicated. **I** AlphaFold3-predicted structure of WT AraC-LOV and comparison to that of R2-LOV carrying the SG to K left linker substitution. A full assessment of model confidence, including pLDDT and PAE metrics is provided in Supplementary Fig. 21. **J** Functional validation of enriched R2-LOV linker variants using the RFP reporter assay. Samples induced with 400 µM IPTG were incubated in the dark or illuminated with blue light. RFP fluorescence and $OD_{600}$ were measured in a plate reader. Data shown as mean ± S.E.M. from $n = 3$ biological replicates.

trigger (Fig. 5G and Supplementary Fig. 16A, B). A distinct enrichment pattern emerged, indicating strong selective pressure favoring specific linker compositions—a phenomenon consistently observed across both replicate pools (Fig. 5G and Supplementary Fig. 16C, D). Over 90% of enriched variants carried mutations in at least one linker. Surprisingly, more pronounced enrichment was observed at the "left" junction, dominated by the replacement of the original SG motif with a single lysine, which accounted for over 40% of variants. In contrast, C-terminal (right) linker mutations were less common, appearing in only ~17% of sequences (Fig. 5G, H, Supplementary Fig. 16C–E).

Sequencing of individual plaques confirmed these trends (Supplementary Fig. 16C), revealing a striking convergence on specific amino acid substitutions within the left linker. Arginine (R), lysine (K), glutamine (Q), and asparagine (N) consistently replaced the original serine-glycine motif within the left linker, suggesting selective pressure for side chains with hydrogen-bonding or helix-stabilizing potential. AlphaFold3[51] structural predictions suggested that the lysine substitution extends the left linker into a continuous α-helix that seamlessly connects the LOV domain to the AraC effector (Fig. 5I). The observed helix extension likely enhances allosteric transmission in a particularly dynamic protein region by minimizing structural discontinuities at the sensor-effector interface, providing a potential mechanistic rationale for the strong enrichment of these left linker variants specifically upon InDel mutagenesis and selection. To validate its functional impact, we cloned five highly enriched variants into an IPTG-inducible plasmid with a weak RBS as done before. All variants showed very strong activity in the dark and practically digital switching upon illumination, consistent with improved transmission of conformational signals from the LOV domain to AraC (Fig. 5J).

Next, we asked whether the most enriched linker mutation would also enhance switching when transplanted into WT-LOV and the lead R2-based AraC-LOV variants resulting from POGO-PANCE (see Fig. 4). Introducing the highest enriched variant, the LdelK linker mutation, into WT-LOV increased dark-state activity approximately 8-fold in the presence of 10 mM L-arabinose, resulting in an improved (>100-fold) dynamic range of light control (Supplementary Fig. 17A). In contrast, incorporation of LdelK into the R2-1.1.3 and R2-2.1.3 backgrounds either resulted in high, but constitutive activity (R2-1.1.3) or strongly impaired activity (R2-2.1.3) (Supplementary Fig. 17B), indicating that the mutational context of the DP6-evolved variants is incompatible with maintaining switching behavior when the LdelK-associated structural changes are introduced. These results suggest that the evolutionary solutions enabled by InDel diversification of receptor–effector junctions are not necessarily compatible with those obtained through DP6-based point mutagenesis (see "Discussion").

Having established that the evolved variants differ in how they accommodate structural changes at the linker, we finally asked whether perturbing promoter architecture would reveal additional mechanistic differences that emerged during evolution of the AraC–LOV fusions. To investigate this possibility, we tested WT-LOV, R2-LOV, R2-LOV LdelK, R2-LOV 1.1.3, and R2-LOV 2.1.3 on promoters in which either the $O_2$ or $I_2$ half sites were replaced with a random, GC-content matched sequence (the −35 element within $I_2$ was preserved) (Supplementary Fig. 18A). Scrambling $O_2$ generally preserved dark-state activity and light-dependent switching across constructs (Supplementary Fig. 18B–F). Fold switching even increased from ~81x to 143x in WT AraC-LOV (Supplementary Fig. 18b), from 454x to 550x in R2-LOV LdelK (Supplementary Fig. 18D), and from 56x to 86x in R2-LOV 1.1.3 (Supplementary Fig. 18E). In contrast, scrambling $I_2$ collapsed dark-state activity resulting in impaired switching in WT AraC-LOV (1.4x), R2-LOV (1.9x), and R2-LOV LdelK (61x) (Supplementary Fig. 18B–D), but markedly enhanced switching in the DP6-evolved variants R2-LOV 1.1.3 and 2.1.3 by decreasing light-state activity (Supplementary Fig. 18E, F). These results suggest that neither $O_2$ looping nor $I_1/I_2$ adjacency is required for switching in the engineered LOV fusions, and that the evolved variants no longer depend on the canonical AraC loop or half-site architecture for activation.

## Discussion

Our work introduces POGO-PANCE, a phage-assisted evolution platform to effectively navigate dual fitness landscapes to evolve switchable protein function. POGO-PANCE retains the broader advantages of PANCE[15,21], including the key benefit of decoupling the evolving protein from host cells. This arrangement eliminates host-genome escape routes and keeps selection focused on the evolving switch, a challenge that has hindered other approaches[52]. We demonstrate the evolutionary power of POGO-PANCE by converting a poorly active, modestly photoswitchable AraC-LOV hybrid into a potent optogenetic transcription factor exhibiting light-regulated control over gene expression spanning more than three orders of magnitude. The mutations that arose during POGO-PANCE evolution suggest that the platform optimized a light-responsive allosteric pathway within AraC, accessed through LOV insertion into R2/R5. The repeated emergence of similar mutations across independent trajectories indicates that this pathway reflects underlying and reproducible principles. This work establishes POGO-PANCE as a platform for dissecting such principles, although defining them fully will require applying the approach to a wider range of receptor–effector architectures.

The success of POGO-PANCE depends on selection steps being aligned with inputs and thus defined functional goals, which previously was observed to be highly challenging in context of Riboswitch evolution[53,54]. Indeed, pools evolved under mismatched regimes failed to yield useful switches due to poor propagation, wash-out, recombination or other types of coding escape mutation (Supplementary Fig. 19A–C), while those exposed to appropriately toggled selection pressures converged on high-performance variants. These outcomes further depended on the stringency and application of both positive and negative selection.

When applying only positive selection on AraC alone, for example, strong constitutive activity emerged and was observed in our RFP reporter assays (Fig. 2F, R2 and R5 variants). This acquisition of constitutive activity did not eliminate arabinose responsiveness, however, as the resulting mutants still displayed arabinose-influenced propagation effects when challenged in our selection circuits (Supplementary Fig. 3B). Their LOV-inserted counterparts also retained arabinose responsiveness while having acquired a light-sensitive activity (Fig. 4B and Supplementary Fig. 5F), indicating that the arabinose and light-dependent properties can collaborate to modulate transcription, presumably through interconnected allosteric networks.

In our DP6-based POGO-PANCE regimes, relatively permissive negative selection likely contributed both to the persistence of R2-/R5-LOV or LOV-excised revertant phage (Supplementary Fig. 19) and to elevated basal leakiness observed in several switches (Fig. 4B). This permissiveness was particularly evident in the R5 background, which overall exhibited reduced light-dependent repression, indicating that the R5-LOV group could have benefited from higher negative selection stringency. Furthermore, the three poorest-switching R5 variants all arose from the R5-P2.2 condition, where positive selection had been applied twice before initiating POGO-PANCE cycling, suggesting that delaying negative selection can hinder enrichment of potent switches when parental variants show very high activity. Tightening repression over time by adjusting the *gIII:gIII-Neg* ratio (Fig. 2H, I) could also be used to more strongly penalize activity in the off-state and thereby enrich lower leaky switches. Together, these observations showcase how selection design shapes which variants persist through POGO-PANCE, setting the stage for understanding how surviving lineages navigate the adaptive landscape.

By integrating deep sequencing into POGO-PANCE, we were able to follow how adaptive trajectories unfolded under opposing selection pressures. This approach provided empirical access to how allosteric communication networks evolve within a defined evolutionary environment[55,56]. The resulting mutational patterns highlight the importance of epistasis and environmental context in shaping these networks, as illustrated by substitutions such as E144K and V223M, which emerged during POGO-PANCE of AraC–LOV but were absent from conventional PANCE of AraC alone (Figs. 2D, 3B). Consistent with this context dependence, linker mutations that were productive in the WT-LOV and R2-LOV backgrounds did not support proper switching in the DP6-evolved variants (Supplementary Fig. 17), indicating that improvements that are advantageous early in evolution can become incompatible once the regulatory landscape has shifted. Moreover, the sustained and even enhanced switching behavior of DP6-evolved AraC–LOV constructs after disruption of key promoter half-sites suggests that variants such as R2-1.1.3 and R2-2.1.3 now rely on regulatory mechanisms that diverge from those used by native AraC (Supplementary Fig. 18).

The distribution of mutations across the three evolutionary regimes further indicates that individual substitutions act through distinct mechanistic routes. R146L consistently behaved as an AraC-centered improvement, suggesting that it enhances regulatory performance of the transcription factor independently of the LOV insertion (Figs. 2F, 4B). In contrast, the highly enriched V223M mutation in the AraC DNA-binding domain repeatedly arose during AraC-LOV evolution (Fig. 3B) but not during the initial PANCE of WT AraC (Fig. 2D), suggesting that it helps compensate for or refine the structural effects of LOV insertion. The repeated appearance of E144K alongside V223M suggests that these substitutions function within a compatible adaptive framework, whereas the rarity of combinations such as V223M with S225R points to mutually exclusive solutions that lead the protein toward different fitness peaks (Supplementary Fig. 9B). Taken together, these patterns indicate that the evolving chimera accesses multiple mechanistic routes, with some mutations strengthening core AraC function and others specifically tuning the contributions of the LOV domain even within the original AraC architecture itself.

Crucially, traditional mutagenesis approaches based on MP/DP are highly biased and fall short in exploring the full sequence and structural landscape allosteric coupling, where signal transmission often hinges on subtle geometric features, such as length, helix phasing, or domain orientation. To address this, we developed RAMPhaGE, a retron-based diversification platform that enables precise, programmable introduction of substitutions, insertions, and deletions in a targeted, constrained and cumulative manner. Although M13 replicates via rolling-circle synthesis and lacks a classical bidirectional replication fork[57], our results indicate that the Recombitron functions well in this genomic background. While experimental evidence is still lacking, we hypothesize that editing occurs during the transient double-stranded replicative-form stage[57], i.e., when the incoming ssDNA genome is converted to a duplex that exposes single-stranded windows probably amenable to CspRecT-mediated donor annealing and incorporation.

In the context of allostery, this capacity for editing was particularly important for revealing alternative paths towards optimizing light-dependent behavior. Specifically, while DP6-based POGO-PANCE resulted in mutations favoring the right linker, RAMPhaGE clearly favored alterations of the left linker, enabled by its capacity for deep indel mutagenesis (Fig. 5G–I, Supplementary Fig. 16D, E). In doing so, RAMPhaGE uncovered structural routes and functional solutions that did not occur in DP6-based POGO-PANCE, expanding the evolutionary search space for phage-assisted evolution. In this work, we ran separate evolutions for either DP6-mediated random point mutations or Recombitron-induced linker alterations, which resulted in mutually exclusive evolutionary solutions (Supplementary Fig. 17). An attractive future direction therefore is a hybrid workflow in which retron-mediated InDel and site-focused diversification is followed by DP6 point mutagenesis of the phage pool prior to phage-pool positive and negative selection, thereby searching both the separate and combined sequence spaces for optimal solutions. While the known toxicity associated with MP6/DP6[25] may hinder direct co-expression of DP6 and RAMPhaGE components in a single host, sequential diversification or pooling strains containing the two systems is entirely feasible and would permit co-exploration of both global (DP6) and targeted (RAMPhaGE) sequence spaces. It is worth noting that RAMPhaGE must be seeded at a relatively high MOI because the input titers remain close to the output titers (Supplementary Fig. 11A), whereas with DP6, phage pools can be expanded from titers as low as ~$10^3$ phages/mL. As a result, incorporating a DP6 passage could not only provide additional diversification but also help recover population size before subsequent RAMPhaGE cycles, which may be advantageous for future POGO-PANCE workflows.

Conceptually, POGO-PANCE and RAMPhaGE should be compatible for studying and evolving allostery for a broad range of effector domains as well as inputs, including chemicals, pH or redox state. This opens a path to wide future exploration: Evolving various effectors for trigger-responsive behavior could not only feed applications in biotechnology and biomedicine, but could eventually reveal the fundamental principles underlying allostery. Finally, our platform may open a path toward the de novo evolution of switchable proteins. While this remains a formidable challenge, analogous to how nature repeatedly evolved photosensitive proteins from unrelated scaffolds, POGO-PANCE provides the evolutionary infrastructure for similar efforts in the lab.

While we recognize the challenge associated with finding suitable sites for receptor fusion and allosteric regulation—a prerequisite for POGO-PANCE—, the recently developed ProDomino machine-learning pipeline from our group simplifies this process[9]. Sampling a small number of top-ranked, ProDomino-inferred sites via receptor insertion therefore provides a pragmatic strategy to obtain an initial, switchable scaffold suitable for subsequent POGO-PANCE evolution.

In summary, the combined POGO-PANCE and RAMPhaGE framework enables not only the directed evolution of powerful protein switches, but allows us to spy into the evolutionary trajectories that yield well-orchestrated allosteric networks. Together, they provide a powerful foundation for both practical applications of and fundamental insights into allostery.

## Methods

### Plasmid, M13 Phage, and strain preparation

Plasmid sequences, strains and selection phages are listed in Supplementary Data 1, primer sequences in Supplementary Data 2. Unless otherwise noted, all *E. coli* strains (TOP10, S2060[34], S2208[16,19]) used for cloning or downstream experiments were transformed with plasmids and plated on LB agar with the appropriate antibiotic concentrations: ampicillin (100 µg/mL, Sigma, cat. no. A9518), kanamycin (50 µg/mL, Applichem, cat. no. A1493), or chloramphenicol (25 µg/mL, Applichem, cat. no. A7495). Colonies were picked and grown in LB medium with selection antibiotics overnight at 37 °C and shaking at 220 rpm. For downstream experiments, glycerol stocks were prepared from single colonies by mixing cultures 1:1 with 50% glycerol and storing in cryovials (or PCR tubes for the S2060 retron editing libraries) at −80 °C.

All media and agar-based products, including LB, 2xYT agar, and top agar, were prepared in-house. Standard agar formulations (LB and 2xYT agar) were made using bacteriological agar at 12 g/L (Applichem, cat. no. A0949). LB was composed of 5 g/L yeast extract (Gibco, cat. no. 9070604), 10 g/L tryptone (Gibco, cat. no. 211705), and 5 g/L NaCl (Applichem, cat. no. 381659.1211). 2xYT was prepared at 31 g/L using Sigma medium (cat. no. Y2377). Soft agar overlays (-0.8% agar) for plaque assays were prepared using 20 mL molten 2xYT agar and 30 mL 2xYT medium, supplemented with X-gal (40 mg/mL stock, Roche, cat. no. 3117073001).

All plasmids were amplified with primers ordered from Merck and assembled using one-pot Golden Gate reactions[58]. PCR fragments were amplified using Q5 Hot Start High-Fidelity DNA Polymerase (New England Biolabs, cat. no. M0493S). PCR products were gel-purified and assembled using NEB Type IIs restriction enzymes and T4 DNA Ligase (Thermo Fisher Scientific, cat. no. EL0011). For assemblies not utilizing plasmid DNA, DpnI (New England Biolabs, cat. no. R0176S) was included to degrade methylated template DNA. Type IIs enzymes used included Esp3I (New England Biolabs, cat. no. R0734S), BsaI-HF®v2 (R3733S), BbsI (R0539S), SapI (R0569S), and PaqCI® (R0745S). Plasmids were assembled via Golden Gate cloning.

Plasmids were purified using the QIAprep Spin Miniprep Kit (Qiagen, cat. no. 27104) or, for very low-copy origin plasmids, the ZymoPURE Plasmid Miniprep Kit (Zymo Research, cat. no. D4210). Constructs were verified by Sanger sequencing using the Economy Run single-tube service (Microsynth).

All phage-based protocols were adapted from those described in Miller, Wang, and Liu[15]; readers should defer to this publication for additional methodological details related to phage assembly, handling, production, and details not explicitly outlined in this "methods" section. M13 phage genomes lacking *gIII* were assembled using M13 backbone vectors and *ccdB*-based entry plasmids for inserts and Golden Gate reactions were transformed into chemically competent *E. coli* S2208 (a S2060 derivative harboring a PSP-driven *gIII* cassette and a *LacZ* cassette on the F-plasmid) and directly added to 2xYT medium with ampicillin at 37 °C and 220 rpm shaking for overnight growth. Cultures were centrifuged and the supernatant filtered through Filtropur S PES 0.2 µm syringe filters (Sarstedt AG, cat. no. 83.1826.001) using 1 mL BD Plastipak Tuberkulin syringes (Becton Dickinson, cat. no. 303172). Phage clones were isolated by plaque assay, Sanger-sequenced, and propagated for downstream experiments (see Phage Quantification and Molecular Analysis section for details).

### mRFP reporter assays

For pBAD-*mRFP* reporter assays, plasmid encoding an AraC variant and a pBAD-driven *mRFP* reporter (Supplementary Fig. 1E) were co-transformed into *E. coli* TOP10 cells. Cultures were inoculated from glycerol stocks and grown overnight as precultures at 37 °C with 220 rpm shaking. The following day, bacteria were diluted 1:50 into 96-well plates (CytoOne, cat. no. CC7672-7596) containing 200 µL of LB medium supplemented with appropriate antibiotics (Kanamycin and Chloramphenicol). Chemical inducers included L-arabinose (Roth, CAS 5328-37-0) and IPTG (Roth, cat. no. 2316.4) as indicated. Cultures were incubated for 18 h at 37 °C with 220 rpm shaking. Reporter activity (RFP fluorescence) and cell density ($OD_{600}$) were measured using a Tecan plate reader (Tecan, Tecan Infinite 200 Pro) with an excitation wavelength of 585 nm and an emission wavelength of 620 nm. Fluorescence values were normalized to $OD_{600}$.

Each condition was performed in three technical replicates per plate. Three biological replicates were collected on separate days using independently grown precultures inoculated from the same glycerol stock. Unless otherwise noted, individual data points correspond to the mean of three technical replicates.

For reporter assays involving optogenetic protein variants, blue light stimulation was applied to one plate during incubation using custom blue light arrays (see "Blue Light Illumination Setup" section), while a parallel control plate was incubated in the dark in a separate incubator under the same temperature and shaking conditions for 18 h.

### Flow cytometry

mRFP reporter assays were performed as described above. After 18 h of culture growth, 2 µL of each sample were diluted into 200 µL of sterile 1x PBS (Roth, cat. No. 9150.1) in a 96-well flat-bottom plate (Greiner, polystyrene, cat. no. M2936) and analyzed using a CytoFLEX flow cytometer (Beckman Coulter, CytoFLEX S) operated at medium flow rate. For each sample, 20,000 events were recorded. Forward scatter (FSC) and side scatter (SSC) were used to gate single *E. coli* (see gating strategy in Supplementary Fig. 2). Fluorescence was measured using the yellow laser and a 610 nm emission filter. Data were analyzed using the CytoFlow Python package.

### Blue light illumination setup

For mRFP reporter assays and flow cytometry experiments, blue light was applied using a custom-built 470 nm LED device utilizing a previously described setup[36], consisting of eight high-power LEDs (CREE XP-E D5-15, LED-TECH.DE) mounted to an aluminum plate and powered by a switching-mode power supply (Manson HCS-3102). The array was installed upside down inside a shaking incubator (Eppendorf, model EPM1335-0000), approximately 5 cm above 96-well sample plates, and operated continuously at an intensity of 5.5–5.9 W/m². A matching control plate was housed in a light-shielded incubator under identical temperature (37 °C) and shaking (220 rpm) conditions. A photograph of the setup is shown in Supplementary Fig. 5A.

For phage evolution experiments, optogenetic stimulation was provided by adhesive 460-465 nm LED strip lights (YUNBO, B50W1-B) affixed to plastic test tube holders mounted at the center of the metal rack inside an orbital shaker incubator (INFORS HT, Minitron). The strips were powered by a constant voltage power supply (Ledmo, KW805) and positioned with two stripes to illuminate the shaking platform from a side angle and ensure full exposure of all culture vessels. A photograph of the phage evolution light configuration is shown in Supplementary Fig. 5B. Light intensity at the surface of plate/tube contact in both experimental setups were measured using a calibrated photodiode power meter (LI-COR, LI-250A) and set to a range from 5.5 to 5.9 W/m². Blue light was applied continuously during the designated light phase of each experiment, with detailed timing

and conditions provided in the corresponding figure legends or protocol sections.

## Phage-based experiments

For all phage-based experiments, *E. coli* S2060[34] strains transformed with *gIII*-containing circuit plasmids were used. Strains containing selection circuits were maintained as precultures inoculated from glycerol stocks, grown overnight at 37 °C with 220 rpm shaking, stored at 4 °C, and (during evolution experiments) refreshed weekly as necessary. Fresh cultures were initiated daily by diluting precultures 1:100 into LB medium and grown to mid-log phase prior to phage infection and inducer addition.

Two exceptions were applied to this standard workflow. *E. coli* cells transformed with the pLlacO1-DP6 MP were freshly transformed each week to preserve induction fidelity, grown directly to mid-log phase, and stored at 4 °C for short-term use. To assess mutagenic activity, individual colonies were plated on LB agar half-plates with or without 1 mM IPTG and incubated overnight at 37 °C to confirm mutagenesis activity through IPTG-dependent lethality. For retron library experiments, *E. coli* S2060 was first electroporated with library plasmids, grown overnight in selective media, and aliquoted into 200 μL volumes containing a 1:1 mixture of LB and 50% glycerol in PCR tubes. These were stored at −80 °C, and for each experiment, one aliquot was thawed and used in full to initiate an overnight preculture (see "Retron Library Preparation" for additional details).

Unless otherwise noted, all experiments were initiated by infecting mid-log-phase cultures with $1 \times 10^6$ plaque-forming units (pfu)/mL of phage. For competition experiments involving two phage variants, $5 \times 10^5$ pfu/mL of each were added. Chemical inducers were added at the time of infection. Specific circuit configurations, induction schedules, and selection regimes are described in the corresponding figure panels and legends or found in the annotations in Supplementary Data 1 and 3–5.

## Phage titer quantification and molecular analysis

Phage titers were primarily determined by plaque assay, with SYBR Green-based qPCR performed during select evolution experiments as indicated. For plaque assays, phage samples were serially diluted in nuclease-free water in 100-fold increments, mixed with mid-log-phase *E. coli* S2208 host cells (OD$_{600}$ ≈ 0.3−0.6) grown on the same day, and combined with molten 2xYT top agar (~0.8% agar). The mixture was plated into 12-well plates (Corning, cat. no. CLS3737) and incubated overnight at 37 °C. Blue plaques (from PSP-*LacZ* cassette on F- plasmid) were picked for colony PCR using Phusion Flash High-Fidelity PCR Master Mix (Thermo Fisher Scientific, cat. no. F548S), and amplicons were Sanger sequenced (Microsynth).

For downstream use as monoclonal phage isolates, sequence-verified plaques were picked, propagated in liquid culture at 37 °C with 220 rpm shaking, centrifuged to pellet host cells, and the resulting supernatant filtered through Filtropur S PES 0.2 μm syringe filters.

For qPCR-based phage titration, samples were first heat-inactivated, treated with DNase I (cat. no. M0303S, New England Biolabs), and used directly as template in 10 μL reactions using the PowerUp SYBR Green Master Mix (cat. no. A25742, Thermo Fisher Scientific) and the primers listed in Supplementary Data 2. PCR was performed on a Quantabio Q qPCR instrument (Quantabio, cat. no. 95900-4 C) and analyzed using Q-qPCR Software v1.0.2. Phage titers were determined by comparison to a standard curve generated from plaque-assay-quantified phage samples. Phage populations were also periodically assessed by "Pooled" PCR reactions of phage containing supernatant and Sanger sequencing (Microsynth) of resulting amplicons to monitor the enrichment of dominant variants or retron editing outcomes. For quantification of retron editing experiments, ~10,000 phage genomes were sampled based on estimated titers.

## Phage assisted non-continous evolution (PANCE)

PANCE[17] was used to evolve constitutively active AraC variants from a monoclonal M13 phage encoding wild-type AraC. Six parallel phage pools were subjected to repeated cycles of mutagenesis and positive selection under either L-arabinose-free or L-arabinose-gradient conditions, as outlined in Fig. 2C.

Each cycle began by infecting freshly transformed *E. coli* S2060 cells carrying the pLlacO1-DP6 MP with ~10³ pfu/mL in 5 mL LB medium supplemented with chloramphenicol, 1 mM IPTG, and 50 ng/mL anhydrotetracycline (aTc; Merck, 137919). Cultures were grown overnight at 37 °C with 220 rpm shaking to induce mutagenesis. The following day, phage-containing supernatants were harvested by centrifugation and filtration through Filtropur S PES 0.2 μm syringe filters, and titers were quantified by qPCR. Filtered phages were then used to infect mid-log-phase S2060 cells carrying the accessory plasmid for positive selection (AP$_{Pos}$). L-arabinose was simultaneously added where appropriate based on the selection regime. Cultures were incubated overnight in LB medium with ampicillin at 37 °C with 220 rpm shaking to amplify phage, and supernatants were collected for the next round. Five total mutagenesis-selection cycles were performed over 10 days (Fig. 2C).

Phage input titers were adjusted to ~10³ pfu/mL during mutagenesis to maximize mutation load and to ~10⁶ pfu/mL during positive selection to maintain diversity. The final round of positive selection included a dilution to ~10³ pfu/mL to impose a stringent, activity-dependent bottleneck prior to clonal isolation (Supplementary Note 1). Phage propagation was monitored by qPCR throughout. See Supplementary Data 3 for exact details on the selection regimes performed. At the conclusion of evolution, individual plaques were isolated, sequenced with Sanger Sequencing, and subcloned into AraC expression plasmids for functional testing using a dual-plasmid mRFP reporter assay in *E. coli* TOP10 (see "mRFP Reporter Assay" section for details).

## Protein on/off gene optimization using phage assisted non-continuous evolution (POGO-PANCE)

POGO-PANCE was used to evolve light-responsive AraC-LOV hybrids through alternating cycles of negative and positive selection (Supplementary Note 1). In Fig. 3, Two starting variants, R2-LOV and R5-LOV, were selected based on basal activity profiles (Supplementary Fig. 5E, F) and evolved in duplicate across three selection strategies (Fig. 3A).

Each cycle began with mutagenesis in *E. coli* S2060 carrying the pLlacO1-DP6 plasmid. Cultures were inoculated in 5 mL LB medium supplemented with chloramphenicol, 1 mM IPTG, and 50 ng/mL anhydrotetracycline (aTc), and infected with phage at ~10³ pfu/mL. Cultures were incubated overnight at 37 °C with 220 rpm shaking. The following day, phage-containing supernatants were harvested by centrifugation and 0.2 μm filtration, and phage titers were quantified by qPCR. Filtered phages were then carried forward into strategy-defined selection steps.

Negative selection was performed by infecting mid-log-phase *E. coli* S2060 cells harboring the AP$_{Neg}$ and AP$_{Prop}$ plasmids at a 1:100 dilution from the prior step. Cultures were maintained in LB medium with ampicillin and chloramphenicol. At the time of infection, 25 μM IPTG was added to induce low-level wild-type *gIII* expression. Cultures were incubated overnight at 37 °C with 220 rpm shaking in 5 mL volumes within vertical glass tubes, either under continuous blue light illumination or in complete darkness, as defined by the selection strategy (Fig. 3A). Infectious phage were quantified by plaque assay to exclude the possibility of non-infectious pIII-Neg containing phage particles from detection by qPCR.

Positive selection used filtered phage at a titer of ~10⁵–10⁶ pfu/mL to infect mid-log-phase S2060 cells carrying the AP$_{Pos}$ plasmid. Cultures were maintained in LB medium with ampicillin and incubated overnight at 37 °C with 220 rpm shaking in 5 mL volumes within

vertical glass tubes. During incubation, cultures were either illuminated with continuous blue light or kept in complete darkness, as defined by the selection strategy (Fig. 3A). Infectious phage were quantified by qPCR.

To initiate the next cycle, ~10³ pfu/mL of filtered phage from the previous step was used to seed mutagenesis. Phage propagation was tracked by qPCR following mutagenesis and positive selection, and by plaque assay following negative selection. Each selection strategy involved either three or four full rounds over 9–10 days (Supplementary Data 4). Three phage pools (R2 P3.2, R5 P2.1, and R5 P3.1) were excluded from downstream analysis due to poor titer or recombination into LOV-excised pools.

At the conclusion of evolution, four plaques were isolated from each remaining pool. AraC-LOV variants were sequence-verified by Sanger and tested for light-dependent regulation using a dual-plasmid mRFP reporter assay in *E. coli* TOP10 (see "mRFP Reporter Assay" section for details).

In Fig. 5G, POGO-PANCE was performed using R2-LOV as the starting input. The workflow was performed as described above in two duplicate pools (Supplementary Fig. 16A), with the exception that the RAMPhaGE linker library was used for mutagenesis in place of pLlacO1-DP6 using a starting titer of 10⁸ pfu/mL (see "Retron Library Preparation" for more details), 1:100 dilutions were used for each step, and two consecutive rounds of positive selection were applied as final selection steps (Supplementary Data 5).

## Retron library preparation

RAMPhaGE libraries (Supplementary Note 1) were designed for pooled, retron-based mutagenesis of linker and LOV domain regions (Fig. 5C, Supplementary Fig. 12). Oligo pools and barcode primer pairs were generated using the RAMPhaGE_Oligo_Tool.ipynb Jupyter notebook and ordered from Twist Bioscience. The barcode scheme and library cloning strategy were adapted from Coyote-Maestas et al.[59], including the biorthogonal barcoding[60]. Each retron donor encoded a 90 nt sequence with edits centered in the middle of the substituted oligonucleotide.

For the DMS (deep mutational scanning) library, the coding sequence for the LOV domain within the AraC–LOV fusion was systematically targeted. Each amino acid position within the LOV domain was mutated to all 20 possible residues using the most preferred *E. coli* codon for each. No other domain regions were targeted. The complete library can be found in the file "AraC_S170_LOV_DMS_Pool.fasta" in the Zenodo repository (https://doi.org/10.5281/zenodo.15650047).

The linker library encoded variations in the SG (left side) and GS (right side) linker regions flanking LOV at its insertion site within AraC (residue S170). Variants included deletions of 1 to 3 amino acids (denoted del1, del2, del3, respectively) of AraC and the linker at the junctions, and substitutions or insertions of various linkers on the N- and C-terminal sides of the LOV domain. These included single amino acid linkers (X), flexible linkers (e.g., GG, GXG, GGSG, GSGG, GSGSG), and rigid linkers (e.g., PP, GPPG, GPPPG, GPPPPG). Unless otherwise specified, linker variants lacking explicit GS or SG regions were designed to replace the native junction. Additional LOV domain variants were encoded in the library by truncating or extending the Jα helix on the C-terminus using the IDEAAKEL motif from the native *As*LOV2 domain in NPH1-1. These included constructs such as IDEAAKEL, IDEAAKE, IDEAA, IDEA, IDE, ID, and I. The complete library can be found in the files "AraC_LOV_Linkers_(L)_Pool.fasta", "AraC_LOV_Linkers_(R)_Pool.fasta", and "AraC_LOV_Linkers_(R)_Extra_Pool.fasta" in the Zenodo repository (https://doi.org/10.5281/zenodo.15650047); further annotation is provided in Supplementary Data 6.

Retron insert sublibraries were amplified by PCR using Q5 Hot Start DNA polymerase. Products were verified by gel electrophoresis, and desired bands were gel-excised and purified as explained above. Libraries were assembled in Golden Gate reactions and electroporated

into 50 µL of *E. coli* TOP10 (Bio-rad,Gene Pulser Xcell Electroporation System); at 1.6 kV, 25 µF and 200 Ω. Cells were immediately recovered in 450 µL of 2xYT medium and incubated at 37 °C with 225 rpm shaking for 1 h. To assess transformation efficiency, 10 µL of the recovered culture was diluted in 90 µL of 2xYT, and 50 µL of the dilution was plated on selective LB agar. The remaining culture was transferred to 50 mL of 2xYT supplemented with chloramphenicol and grown overnight at 30 °C. Colony counts confirmed >7000 transformants (>45× coverage), and three colonies were picked and Sanger-sequenced to verify library integrity. Of note, the RcP9 backbone includes a *ccdB* dropout cassette for negative selection during library cloning.

Validated plasmid libraries were electroporated into *E. coli* S2060 using the same electroporation conditions as described above. Transformed cultures were grown overnight in 2xYT medium supplemented with 25 µg/mL chloramphenicol at 30 °C and stored as 200 µL aliquoted glycerol stocks at −80 °C. Each aliquot was used in full to initiate a preculture for pooled evolution experiments or to assess plasmid input library coverage via Illumina next-generation sequencing.

Complete sequences of all library variants, including DMS and linker retron donors, are provided in their corresponding FASTA files, with barcode and primer associations listed in Supplementary Data 2. Nomenclature for linker variants used in Fig. 5 and related panels is defined in Supplementary Data 6. For additional information on oligo pool design and targeting strategy, please refer to annotated code on GitHub (https://github.com/Niopek-Lab/POGO_PANCE.git) and to Supplementary Fig. 12.

## Retron library editing

Retron-mediated editing[47] was used to generate pooled phage variants carrying site-specific mutations encoded by barcoded retron libraries. Editing cycles were performed similarly to standard mutagenesis rounds but used a higher phage input (~10⁸ pfu/mL) to preserve library complexity. To initiate each experiment, a single frozen glycerol aliquot of *E. coli* S2060 harboring the validated RAMPhaGE plasmid library was thawed and used in full to inoculate 10 mL of LB medium with chloramphenicol. Cultures were grown overnight at 37 °C with 220 rpm shaking. The following day, a 1:100 dilution of the overnight preculture was used to seed a fresh editing culture.

At mid-log phase of *E. coli* growth, filtered phage supernatants (~10⁸ pfu/mL) were added along with IPTG (1 mM) and anhydrotetracycline (aTc, 50 ng/mL) to induce retron expression and mutagenesis. Cultures were incubated overnight at 37 °C with 220 rpm shaking. The next day, supernatants were harvested by centrifugation and 0.2 µm filtration, and phage propagation was quantified by qPCR.

For experiments involving sequential application of both deep mutational scanning (DMS) and linker libraries (Supplementary Fig. 15D), phages were passaged once through each library-containing host strain using a higher initial titer (~10¹⁰ pfu/mL) to compensate for possible cumulative titer loss. Resulting phage pools were either subjected to further selection cycles or analyzed by sequencing as described.

## Illumina library preparation and sequencing

Plasmid DNA of the retron editing library was extracted from overnight cultures of *E. coli* S2060 harboring the retron library, using the same glycerol stocks employed for retron editing experiments. Phage genomic DNA was prepared directly from 1 µL of filtered phage supernatant in all cases. Library amplicons were generated with 5' partial Illumina adapter sequences by PCR (Supplementary Data 2) using Q5 Hot Start DNA polymerase (35x cycles) with barcode-indexed primers specific to each sample group (Supplementary Data 2). For the retron library input, 1 µL of 100 ng/µL plasmid DNA was used per 25 µL PCR reaction.

PCR products were verified by agarose gel electrophoresis, gel-excised, and purified in a single column during preparation for multiplexing. Amplicons were normalized to 20 ng/μL, pooled, and submitted to Genewiz (Azenta) for dual-indexed Illumina sequencing (2 × 250 bp paired-end reads) via the EZ-Amplicon service according to their sample requirements.

Phage populations derived from the linker and DMS library mutagenesis were initiated at a starting titer of $1 \times 10^8$ pfu/mL and 50 ng/μL aTc at the onset of the mutagenesis workflow. Similarly, the pLlacO1-DP6 mutagenesis experiment was initiated under standard mutagenesis conditions used during evolution, with a starting titer of $1 \times 10^3$ pfu/mL and 50 ng/μL aTc. These phage populations were subsequently propagated and sampled for Illumina sequencing to quantify editing outcomes or mutational profiles.

Barcode identities are listed in Supplementary Data 2; further information on the data processing, including all scripts employed for data analysis, can be sourced from our Github repository: https://github.com/Niopek-Lab/POGO_PANCE.git; All Illumina datasets consisted of at least ~43,000 forward reads per barcode. Raw sequencing data can be obtained from Zenodo (https://doi.org/10.5281/zenodo.15650047).

## Illumina read processing and analysis

Illumina data were demultiplexed by retaining only read pairs with perfectly matching forward and reverse barcodes to minimize index swapping. To assign reads to the correct samples, a maximum mismatch of 5 bases was allowed when aligning to the expected primer sequences, accommodating mutagenesis events at primer sites. Reads were trimmed at the first base with $Q < 20$, and adapter and primer sequences were removed. For retron input libraries, only forward reads were used due to reduced quality and positional dropout in reverse reads.

Reads were aligned to reference sequences using BLASTN with default parameters, with the exception that only a maximum of 100k alignments were retained to allow identification of multiple hits. In DMS mutagenesis analyses, reads with indels or shifted alignments were excluded. In linker analyses, in-frame indels were allowed (net indel length divisible by three), and reads were partitioned into linker and insert regions for separate analysis. Only reads spanning the full linker region of interest were retained.

Mutation frequencies were computed at the nucleotide, codon, and amino acid levels by normalizing variant counts to the total read depth per position. Positions with fewer than 2,000 reads were excluded from downstream analysis. For retron input libraries, only forward reads were used for analysis due to lower quality and potential positional dropout in the reverse reads.

For the pie charts in Fig. 5D, forward and reverse reads were aligned to the reference sequence and paired based on read IDs. A read pair was classified as wild-type only if both reads showed zero mismatches to the reference. If either read contained a mismatch, the pair was considered a mutant. Pairs with a combined alignment length less than 80% of the reference were excluded.

Position 446 (G → A) was excluded from nucleotide level analysis in Supplementary Fig. 13C and Supplementary Fig. 14B due to a high background mutation rate (~95%), likely resulting from an unannotated silent mutation or early off-target propagation event rather than a library-encoded substitution.

For codon-level mutation complexity analysis (Supplementary Fig. 13C), each observed codon substitution in the LOV region was categorized as a single-, double-, or triple-base change based on alignment to the WT sequence. Observed frequencies were computed from phage-derived Illumina reads. Expected frequencies were determined by categorizing all codon substitutions encoded in the retron DMS library according to the number of base changes required, assuming uniform library coverage and editing efficiency across all library variants.

In the plots comparing retron library inputs versus encoded phage editing outcomes (Fig. 5F, Supplementary Fig. 13D, Supplementary Fig 14B, Supplementary Fig. 15B), the frequency of encoded DMS mutations for each nucleotide position was calculated as the total count of encoded changes per position, normalized by the total count of encoded mutations across all positions. Similarly, phage-edited substitution frequencies were normalized by dividing each position's mutation frequency by the sum of all position frequencies.

Illumina-based data were used in Fig. 5D–F, H; Supplementary Fig. 13A–E; Supplementary Fig. 14A, B; Supplementary Fig. 15A, B, D; see Supplementary Data 2 and the Github Repo (https://github.com/Niopek-Lab/POGO_PANCE.git) for a more detailed breakdown.

## Nanopore library preparation and long-read sequencing

Phage DNA libraries derived from POGO-PANCE evolution, RAMPhaGE evolution, and retron-mediated editing experiments were sequenced using the Oxford Nanopore Technologies (ONT) MinION platform with FLO-MIN114 flow cells and the Native Barcoding Kit 24 V14 (SQK-NBD114.24, ONT). Library preparation followed ONT's ligation-based amplicon workflow, using barcodes NB01–NB24 across two sequencing runs (Supplementary Data 2).

Phage DNA amplicons were generated as described above. Amplicons were gel-purified and quantified using a NanoDrop spectrophotometer (Implen, NP80 Touch). Only samples with OD260/280 < 1.8 and OD260/230 between 2.0–2.2 were used for downstream processing, in accordance with ONT's quality guidelines.

For barcode ligation, 200 fmol of input DNA per sample was calculated based on amplicon length (e.g., ~130 ng for a 1 kb product). End repair and A-tailing were performed using the NEBNext Ultra II End Repair/dA-Tailing Module (New England Biolabs, cat. no. E7546S), followed by barcode adapter ligation with the NEB Blunt/TA Ligase Master Mix (New England Biolabs, cat. no. M0367L). Barcoded samples were pooled, purified with AMPure XP beads, and ligated to sequencing adapters using the NEBNext Quick Ligation Module (New England Biolabs, cat. no. E6056S). Final libraries were purified using the Short Fragment Buffer provided in the ONT kit. Sequencing was performed on two FLO-MIN114 flow cells using the MinION Mk1B platform (ONT, MIN-101B). Raw sequencing data can be obtained from Zenodo (https://doi.org/10.5281/zenodo.15650047).

## Nanopore read processing and analysis

Nanopore data were processed using Oxford Nanopore Technologies' (ONT) Dorado basecaller with super high-accuracy settings ($Q \geq 10$). After demultiplexing based on full barcode concordance, reads were filtered by quality and length using chopper (1800 < len < 2200, average read quality $Q > 20$), and aligned to phage reference genomes using minimap2 with the -ax map-ont preset. Quality control metrics were assessed using NanoPlot on.bam alignments.

Due to the higher indel rate and noise of Nanopore sequencing[61], analysis pipelines differed to Illumina data analysis and between DMS and linker datasets. For DMS analyses, CIGAR string information was used to correct indels and enforce in-frame alignments. Reads were trimmed to a uniform starting point, and only those spanning the region of interest (LOV domain insertion region, excluding linkers) were retained. For linker analyses, indels were explicitly annotated into the read or reference sequence to preserve true biological variation. Linker boundaries were identified using conserved flanking motifs. Reads with net indels not divisible by three were filtered to exclude frameshift variants.

Mutation frequencies were computed at the nucleotide, codon, and amino acid levels by normalizing variant counts to the total read depth per position. Systematic positional biases were observed in the Nanopore base-called data; To account for this, mutation frequency plots across timepoints in the temporal evolution data set in Fig. 3C were corrected by subtracting baseline mutation frequencies

measured after the initial mutagenesis passage (Day 1), thus enabling accurate tracking of true enrichment dynamics. The uncorrected data are shown in Supplementary Fig. 20 for comparison. In the Nanopore endpoint plots (Fig. 3B), co-occurance data (Supplementary Fig. 8, 9B), and logo plots (Supplementary Fig. 9A) position R38 was excluded from the analysis as it was flagged as a consistent systematic error.

Residue-level enrichment in the final-day POGO-PANCE pools in Supplementary Fig. 9 was quantified by computing amino-acid frequencies at each mutated position. For sequence-logo analysis, the fractional enrichment of all amino acids was calculated, and amino acids exceeding 1% across pooled final-day libraries were retained. Amino acids were grouped by physicochemical class for visualization. Pairwise mutation co-occurrence was assessed by tabulating single-site mutation frequencies (blue) and joint occurrences within individual reads (burgundy). Co-occurrence matrices included only mutations present in ≥9.5% of reads for each pool.

For Supplementary Fig. 15C, reads were categorized based on whether linker edits were detected at the left site, the right site, or both. This allowed classification of each variant as wild-type, single-site edited, or double-site edited. Variants with a low frequency (filtered low % variants, noise threshold <0.01%) were excluded to reduce noise from Nanopore Sequencing.

Nanopore sequencing data were used in Fig. 3B–D; Fig. 5G; Supplementary Fig. 7; Supplementary Fig. 8; Supplementary Fig. 9; Supplementary Fig. 11C; Supplementary Fig. 14C; Supplementary Fig. 15C; Supplementary Fig. 16D, E; Supplementary Fig. 20; see Supplementary Data 2 and the Github Repo (https://github.com/Niopek-Lab/POGO_PANCE.git) for a more detailed breakdown.

### AlphaFold3 structural modeling and enrichment mapping
Structural models of AraC-LOV variants were generated using the AlphaFold3[51] web server with default settings and visualized using PyMOL (Schrödinger Inc.). Positional enrichment data, derived from Nanopore-based variant frequency analysis, were mapped onto the structure. A threshold of 0.10 and 0.25, respectively, was applied to highlight significant mutation rates, i.e., fraction of mutated reads, with values above the threshold displayed in the darkest color. Spheres at the α-carbon indicate residues enriched above 1% were colored according to their enrichment. pLDDT, PAE, and interface-confidence metrics is provided in Supplementary Fig. 21.

### Software and statistical analysis
Data analysis was performed in Python (v3.11.1) using the following libraries: pandas and numpy for data manipulation, scipy for statistical analyses, and matplotlib and seaborn for data visualization. Bioinformatic processing of sequencing data incorporated pysam and Biopython, alongside custom Python modules for DNA translation, variant calling, mutation enrichment analysis, and co-occurrence profiling. Flow cytometry data were analyzed using CytoFlow (v1.1.1), with Python (v3.6.7). Structural visualizations of AlphaFold3-predicted models were generated in PyMOL (v3.1.4.1). Plasmid maps were viewed and annotated using SnapGene (Dotmatics). Bar plots were compiled and visualized using GraphPad Prism (v10). The secondary structure annotations in Fig. 3c were generated using a custom Python script, based on secondary structure predictions from PSIPRED (v4.0) via the PSIPRED Workbench webserver and domain annotations for AraC from InterPro. Figures were assembled in Adobe Illustrator. Unless otherwise noted, error bars indicate the standard error of the mean (SEM) from biological replicates ($n = 3$). No samples or data were omitted from the analysis. Spearmen correlation was applied to measure the strength and direction of a monotonic relationship between observed phage mutation frequencies and the underlying composition of retron-encoding plasmid libraries. For both Illumina and Nanopore sequencing, all downstream analysis, i.e., mutation enrichment, positional spectra, co-occurrence profiling, and filtering, was performed using custom Python scripts based on pysam, Biopython, and Jupyter notebook-based visualizations.

### Reporting summary
Further information on research design is available in the Nature Portfolio Reporting Summary linked to this article.

## Data availability
The raw data underlying Figs. 2B, F, 4B, and 5J, as well as Supplementary Figs. S1C, D, S4F, G, S5B, S10A, B, S17A, B, and S18B–F and full gel image in S19B are provided as a Source Data file. Relevant plasmids generated in this work have been deposited in Addgene with downloadable.gbk files containing our plasmid map annotations, and the corresponding Addgene identifiers and all plasmid sequences are listed in Supplementary Data 1. The next-generation sequencing (Illumina and Nanopore) and flow cytometry data generated in this study have been deposited in the Zenodo database under accession code [https://doi.org/10.5281/zenodo.15650047]. All processed data required to interpret, verify and extend the findings of this study are provided in the Supplementary Information, Source Data files, and the GitHub repository archived at the same Zenodo DOI. Materials and additional information that are not included in the above resources are available upon request from the corresponding authors subject to standard material transfer agreements. High-throughput DNA sequencing FASTQ files generated in this study have been deposited in the NCBI Sequence Read Archive under BioProject accession PRJNA1442006. Source data are provided with this paper.

## Code availability
Scripts used for data analysis, RFP measurement data, as well as the tool for RAMPhaGE oligo pool design, are available. The GitHub repository[62] has been archived on Zenodo at [https://doi.org/10.5281/zenodo.15650047]. The Zenodo DOI provides a permanent, versioned reference for the code used in this study.

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

## Acknowledgements

We thank the Niopek and Mathony labs, as well as the thesis advisory committee to N.T.S., Seth Shipman, UCSF and Kerstin Göpfrich, Heidelberg University for helpful discussions. We thank Konrad Herbst for advice on the Nanopore sequencing; Andreas K. Brödel and Mark Isalan for sharing information regarding mutagenesis plasmid; Kevin Esvelt for information regarding media for PANCE; Tobias Stadelmann for support in establishing the PANCE-method in our group. ChatGPT, model 4o and 5.1 was used for language editing. This work was supported by the European Union (ERC, DaVinci-Switches, project number 101041570). Views and opinions expressed are however those of the author(s) only and do not necessarily reflect those of the European Union or the European Research Council Executive Agency. Neither the European Union nor the granting authority can be held responsible for them. D.N. is also grateful for funding by the Aventis Foundation and for funding by the Deutsche Forschungsgemeinschaft (DFG, German Research Foundation) under Germany's Excellence Strategy—EXC-3018/1—53358728. Anna von Bachmann is supported by the Konrad Zuse School of Excellence in Learning and Intelligent Systems (ELIZA) through the DAAD programme Konrad Zuse Schools of Excellence in Artificial Intelligence, sponsored by the Federal Ministry of Education and Research.

## Author contributions

D.N. conceived the study and refined it with N.T.S. and J.M. N.T.S. and D.N. designed experiments. N.T.S. performed experiments with contributions from M.L.G., N.L., A.S.K., and S.W. N.T.S., A.v.B. and A.H. analyzed data. N.T.S., J.M., and D.N. interpreted data. BW created the software tool for RAMPhaGE oligo pool design based on concepts by N.T.S. D.N. supervised the work with support by J.M. N.S. and D.N. wrote the manuscript draft. All authors reviewed and edited the manuscript.

## Funding

## Competing interests

The authors declare no competing interests.
