## [Peer Review file · Nature Communications]

Phage-Assisted Evolution of Allosteric Protein Switches

Corresponding Author: Professor Dominik Niopek

Version 1:

Reviewer comments:

Reviewer #1

(Remarks to the Author)

The authors have comprehensively addressed my comments and the paper is appropriate for publication in my view.

Fig 4B is missing a fold change value for R5-LOV P3.2.1.

(Remarks on code availability)

Reviewer #2

(Remarks to the Author)

My thanks to the authors for their efforts in revising the manuscript in response to my comments. All of my concerns have been addressed and I recommend that the manuscript be accepted. It is very nice work and a strong contribution to Nature Communications.

I noticed two minor errors in Supplementary Note 1, third paragraph. Superscripts are missing in the line "high MOI (108-1010)." I think this should be 10^8 - 10^{10} ? Also in the same paragraph: "we recommend an additional propagation step (using S2208, for example) if phage titers would be too [missing word?] to enter POGO-PANCE selection."

(Remarks on code availability)

Reviewer #3

(Remarks to the Author)

I appreciate the thorough and detailed response to reviews. The addition of new data in S10, S17 and S18 is particularly appreciated. I also appreciate the authors work to make the github code distribution more straightforward to install/run. I have no further comments to address, and congratulate the authors on a very nice manuscript!

(Remarks on code availability)

I have not re-assessed the code since the last round of reviews, but the author's response and the corresponding updates to the github readme / yaml files is appropriate (and helpful).

Referees' comments:

Reviewer #1 (Remarks to the Author)

The authors have comprehensively addressed my comments and the paper is appropriate for publication in my view.

We thank the reviewer for considering our revised manuscript appropriate for publication and for their time reviewing our work.

Fig 4B is missing a fold change value for R5-LOV P3.2.1.(Remarks on code availability)

Thank you for pointing this out, we have added the missing value. We also corrected a minor inconsistency regarding residue numbering for AsLOV2 and corrected this accordingly.

Reviewer #2 (Remarks to the Author)

My thanks to the authors for their efforts in revising the manuscript in response to my comments. All of my concerns have been addressed and I recommend that the manuscript be accepted. It is very nice work and a strong contribution to Nature Communications.

We sincerely thank the reviewer for the thoughtful evaluation of our manuscript, and for the very positive assessment of our work.

I noticed two minor errors in Supplementary Note 1, third paragraph. Superscripts are missing in the line “high MOI (10⁸-10¹⁰).” I think this should be 10⁸-10¹⁰? Also in the same paragraph: “we recommend an additional propagation step (using S2208, for example) if phage titers would be too [missing word?] to enter POGO-PANCE selection.”(Remarks on code availability)

Thank you for pointing out these errors, we have implemented the corresponding text changes.

Reviewer #3 (Remarks to the Author)

I appreciate the thorough and detailed response to reviews. The addition of new data in S10, S17 and S18 is particularly appreciated. I also appreciate the authors work to make the github code distribution more straightforward to install/run. I have no further comments to address, and congratulate the authors on a very nice manuscript!(Remarks on code availability)

We greatly appreciate the reviewer's positive and constructive feedback throughout the review process.

I have not re-assessed the code since the last round of reviews, but the author's response and the corresponding updates to the github readme / yaml files is appropriate (and helpful).